# GeoLLaVA-8K: Scaling Remote-Sensing Multimodal Large Language Models to 8K Resolution

**Fengxiang Wang[1], Mingshuo Chen[2], Yueying Li[1], Di Wang[3,4], Haotian Wang[1],**
**Zonghao Guo[5], Zefan Wang[5], Boqi Shan[6], Long Lan[1], Yulin Wang[5]\*,**
**Hongzhen Wang[5]\*, Wenjing Yang[1]\*, Bo Du[3,4], Jing Zhang[3]\***

[1] College of Computer Science and Technology, National University of Defense Technology, China
[2] Beijing University of Posts and Telecommunications, China
[3] School of Computer Science, Wuhan University, China
[4] Zhongguancun Academy, China [5] Tsinghua University, China [6] Beihang University, China

## Abstract

Ultra-high-resolution (UHR) remote sensing (RS) imagery offers valuable data for Earth observation but pose challenges for existing multimodal foundation models due to two key bottlenecks: (1) limited availability of UHR training data, and (2) token explosion caused by the large image size. To address data scarcity, we introduce **SuperRS-VQA** (avg. 8,376×8,376) and **HighRS-VQA** (avg. 2,000×1,912), the highest-resolution vision-language datasets in RS to date, covering 22 real-world dialogue tasks. To mitigate token explosion, our pilot studies reveal significant redundancy in RS images: crucial information is concentrated in a small subset of object-centric tokens, while pruning background tokens (e.g., ocean or forest) can even improve performance. Motivated by these findings, we propose two strategies: *Background Token Pruning* and *Anchored Token Selection*, to reduce the memory footprint while preserving key semantics. Integrating these techniques, we introduce **GeoLLaVA-8K**, the first RS-focused multimodal large language model capable of handling inputs up to 8K×8K resolution, built on the LLaVA framework. Trained on SuperRS-VQA and HighRS-VQA, GeoLLaVA-8K sets a new state-of-the-art on the XLRS-Bench. Datasets and code were released at GeoLLaVA-8K.

## 1 Introduction

With the rapid development of Earth science, the collection, process, and representation of remote sensing (RS) data have become increasingly important. [1]. Among these data sources, satellite imagery is able to capturing extensive spatiotemporal information about Earth's surface [2, 3], significantly enhancing our geographic understanding of this planet.

Recent advances in multimodal large language models (MLLMs) [4, 5, 6, 7, 8] have significantly improved visual understanding and reasoning, simultaneously facilitating remarkable scientific progress in geoscience for handling remote sensing data [9, 10, 11]. However, despite significant progress of MLLMs across both general and RS domains, current models still fall short in addressing real-world RS tasks, especially in the case of **ultra-high-resolution (UHR) scenarios**. For instance, even leading models like GPT-4o [12] and Qwen-VL series [13] are limited up to 4K resolution, resulting in limited performance (Accuracy < 0.45) on the XLRS-Bench [14], a recent RS evaluation benchmark featuring large image size (e.g., 8K-10K). This gap raises an urgent problem we aim to investigate in this paper:

---

\*Corresponding authors

39th Conference on Neural Information Processing Systems (NeurIPS 2025).

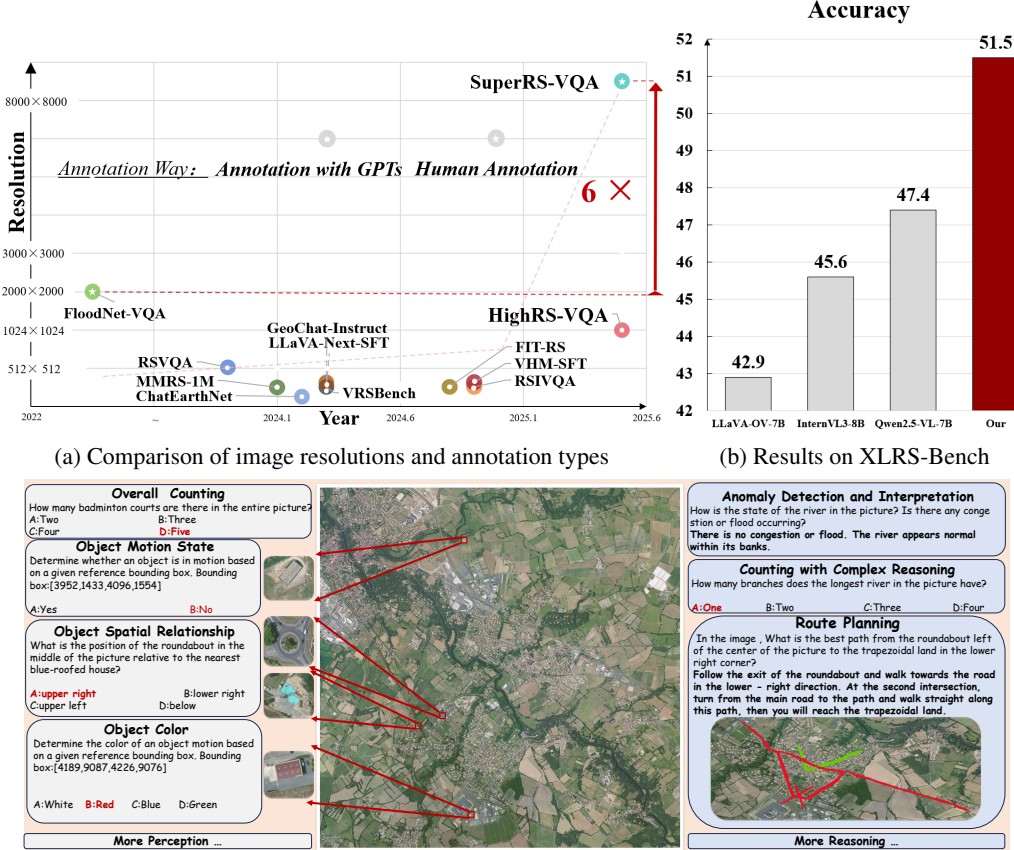

*Can we develop MLLMs tailored for RS data with resolutions far beyond current limits (e.g., up to 8K×8K)?*

(a) Comparison of image resolutions and annotation types

(b) Results on XLRS-Bench

(c) Example of our dataset

Figure 1: We introduce SuperRS-VQA and HighRS-VQA, the highest-resolution VQA datasets for MLLM training. GeoLLaVA-8K, trained on these datasets, significantly outperforms existing MLLMs on ultra-high-resolution remote sensing tasks.

In response to this question, we identify two critical challenges that remain underexplored: **(1) The lack of image-text training data for UHR RS images; (2) The massive number of visual tokens in UHR RS imagery increases training difficulty for MLLMs.** To address the first issue, we introduce two novel multimodal RS datasets featuring large image sizes: SuperRS-VQA (about 8K×8K) and HighRS-VQA (about 2K×2K). To our knowledge, they are the largest image-size RS vision-language datasets to date, covering 22 real-world subtasks and significantly surpassing previous RS image-text datasets in both scale and diversity.

For the second problem, from our intuition, the excessive sequence length of visual tokens may lead to two major issues in MLLM's training: **(1) Expensive Computation Overhead:** Current models are not designed to operate at such scales. For example, directly adapting LLaVA to 8K×8K inputs results in memory overflow. **(2) Low Semantic Density:** UHR RS imagery contains an abundance of homogeneous background tokens, which contribute little useful information. In contrast, semantically rich foreground tokens are sparse and risk being overlooked without effective token management.

To tackle these challenges, we propose two token compression strategies: Background Token Pruning and Anchored Token Selection, which focus on token aggregation and refinement to efficiently manage the large number of visual tokens introduced by UHR inputs. Building on these techniques, we develop GeoLLaVA-8K, a UHR-oriented, RS-specific MLLM capable of processing inputs up to 8K resolution. Experiments on large image-size benchmarks (e.g., XLRS-Bench) demonstrate that our model outperforms all existing open- and closed-source MLLMs, setting a new state-of-the-art.

In summary, our main contributions are as follows:

(1) We curate SuperRS-VQA and HighRS-VQA, two RS image-text datasets covering 22 real-world subtasks for UHR scenes, featuring so far the largest image sizes in our knowledge.

(2) We investigate the challenges in training MLLMs caused by the excessive visual tokens in UHR RS imagery. To address the problems of redundant backgrounds and the sparsity of semantically rich objects, we propose two strategies for token pruning and selection.

(3) We develop GeoLLaVA-8K, the first RS MLLM tailored for UHR scenes, capable of processing inputs up to 8K. Experiments on representative UHR RS benchmarks demonstrate its superior performance compared to existing open- and closed-source MLLMs.

## 2   Related Work

**Remote Sensing Multimodal Datasets.** With the rapid progress of large multimodal models in general domains, the RS field has witnessed significant advancements in MLLM development [9, 10, 11, 15, 16], leading to the emergence of several RS-specific vision-language datasets. RSVQA [17] comprises image/question/answer triplets, with questions and answers derived from OpenStreetMap (OSM). RSIVQA [18] generates samples automatically using existing scene classification and object detection datasets. RSSA [19] targets hallucination, while FIT-RSRC [16] focuses on object relationship understanding. VRSBench [20] includes 29,614 images, 52,472 object references, and 123,221 QA pairs. Recent datasets for the SFT stage of MLLM training include GeoChat_Instruct [9], ChatEarthNet [21], VHM_sft [22], FIT-RS [16], and MMRS-1M [15]. However, most of these datasets are aggregated from existing sources and contain limited original annotations. Critically, their average image resolution remains below 1K×1K, which is insufficient for real-world UHR RS tasks.Recently, XLRS-Bench [14] introduces the largest image-size RS benchmark to date (average 8,500×8,500) for evaluation purposes only, underscoring the persistent lack of corresponding UHR training data in the RS MLLM field.

**Multimodal Large Language Model.** Leveraging advanced large language models (LLMs) like GPTs [5] and LLaMA [6], MLLMs have demonstrated strong visual understanding and reasoning capabilities [23, 24]. Proprietary models such as Gemini [4] and GPT-4o [5], along with open-source alternatives like Qwen-VL [8], InternLM-XComposer [25], MiniCPM [26], LLaVA [27], and MiniGPT-4 [7], show competitive performance but are typically limited to input resolutions of 2K-4K. To overcome this limitation, LLaVA-Next [28] processes images in patches and connects them via global tokens, Monkey [29] and LLaVA-UHD [30] compress patches to reduce redundancy, Cambrian [31] uses learnable queries for multi-scale interaction, and SliME [32] applies dual compression to preserve global context. In the RS domain, several MLLMs have been developed. For example, GeoChat [9] enables multi-task RS dialogue based on the LLaVA-1.5 framework [33]. EarthGPT [15] unifies multisensor interpretation tasks by converting RS annotations into question-answer pairs. Meanwhile, LHRS-Bot [10] improves vision-language understanding through multi-level alignment and curriculum learning. Nevertheless, both general-domain and RS MLLMs struggle to effectively handle UHR RS imagery. Universal MLLMs lack domain adaptation and are unable to meet the higher resolution requirements of real-world RS applications. Although RS-specific MLLMs incorporate domain knowledge, they still operate at significantly lower resolutions, i.e., less than 1K, highlighting the urgent need for a UHR (8K×8K) and domain-aligned MLLMs in the RS field.

## 3   Dataset

The RS domain currently lacks sufficient UHR image-text training data for MLLM development. To validate this judgment, we firstly unified existing RS image-text datasets with general-domain data in a 2:1 ratio for supervised fine-tuning (SFT). However, we observed a notable performance drop in LLaVA-Next-2K [34] on UHR benchmarks (XLRS-Bench [14]) in the Tab. 1, which we attribute to the limited resolution of current RS training datasets, highlighting the urgent need for UHR image-text pairs to support real-world RS applications.

To address this, following XLRS-Bench [14], we first manually annotated 12K UHR samples. Notably, existing MLLMs such as GPT-4o [12] either encounter memory overflow or generate low-quality outputs when processing UHR RS imagery. Therefore, we adopt a fully manual annotation

approach in this work. Nevertheless, due to the high time and labor costs, scaling manual annotation of UHR data is impractical. To address this, we develop a semi-automated annotation pipeline that leverages GPT-4o [12] alongside existing RS detection and segmentation datasets, generating 100K medium-to-high-resolution (MHR, 2K×2K) image-text pairs. To bridge the distribution gap between MHR and UHR data, we further apply an influence-based data selection method built on the LESS framework [35]. Next, we detailed the procedure of constructing datasets.

**Source Screening.** Our datasets have diverse and extensive image sources. We follow a key principle: to avoid using the same data sources as existing vision-language datasets as far as possible, ensuring diversity and effectiveness in training data. Specifically, we utilize datasets such as Deep-Globe [36], STAR [37], FAIR1M 2.0 [38], LoveDA [39], Inria [40], OpenSatMap [41], HRSCD [42], MiniFrance [3], and DOTA [2]. To minimize redundancy of image-text pairs, we deduplicate images within these datasets and remove overlaps with existing benchmark datasets like XLRS-Bench [14]. Nonetheless,

Table 1: **Pilot Experiments of Dataset.** We keep the original pretraining data of LLaVA-Next-2K [34] (a variant of LLaVA-Next [28]) and we only modify the SFT (supervised fine-tuning) data in the second training phase. Accuracy is reported based on results from XLRS-Bench [14].

| Model | SFT Data | | | Acc. |
|---|---|---|---|---|
| | Data Source | Volume | Average Resolution | |
| LLaVA-Next | LLaVA-Next-2K_SFT [28] | 738K | 512×512 | 44.6 |
| LLaVA-Next | LLaVA-Nex-2K_SFT [28] | 738K | 512×512 | 40.4↓ |
| | GeoChat_Instruct [9] | 283k | 632×619 | |
| | ChatEarthNet [21] | 6k | 256×256 | |
| | VHM_sft [22] | 150k | 643×638 | |
| | FIT-RS [16] | 1200k | 512×512 | |
| | MMRS-1M [15] | 308k | 499×491 | |
| | VRSBench-train [20] | 85k | 512×512 | |

it should be noted that, since generating MHR image-text pairs requires leveraging existing annotations with tools like GPT-4o, the overlap with prior VQA datasets is unavoidable.

**Annotation.** For UHR data, we formed a team of 5 MLLM and RS experts (Ph.D. holders or candidates) and 30 crowd-sourcing annotators (undergraduate and masters students). All samples were manually labeled and rigorously cross-validated over 40 days of annotation and 10 days of verification, resulting in a 12K UHR image-text dataset, named SuperRS-VQA. For MHR data, we developed a semi-automated annotation. Using task-specific prompts and existing annotations (e.g., bounding boxes in RS detection datasets), we generated text via GPT-4o [12]. Despite high token costs (>$1,000), outputs still needed to be quality-checked by annotators.

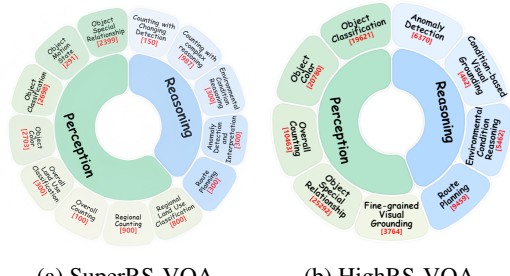

(a) SuperRS-VQA     (b) HighRS-VQA

Figure 2: Our datasets have the perception and reasoning dimensions across 22 sub-tasks.

**Data Selection Pipeline for MHR Data.** We further adopted an influence-based data selection pipeline (see Fig. 3) to improve the relevance of our dataset to UHR downstream tasks and ensure its cultivation of reasoning capabilities for models fine-tuned on it. This pipeline quantifies the exact contribution of each training data sample by measuring the change in validation performance when that sample is removed (i.e., a Leave-One-Out influence score). Concretely, we leverage gradient information from a warm-up fine-tuned model $\theta$ trained on the crude dataset: for each candidate $z \in Z$ among training examples and validation examples $x \in X$, we compute

$$\text{Inf}_{\text{SGD}}(z_i, X) = \max_{x \in X} \cos\langle \nabla \ell_\theta(z_i), \nabla \ell_\theta(x_i) \rangle \tag{1}$$

where $\ell_\theta$ is the negative log-likelihood loss on a data sample. The above estimation is a simplified but empirically valid form of *Influence Function* [43], a widely used data valuation method which is a first-order approximation of performance change. Intuitively, a higher $\text{Inf}_{\text{SGD}}$ indicates removing $z_i$ from the training set would cause a larger, more directionally coherent shift in validation loss, and hence that $z_i$ is more influential. Unlike heuristic selection methods based on the apparent semantics of VQA pairs, which often require laborious task-specific rule engineering, our approach directly measures data influence utilizing the preferences of a fine-tuned model.

We follow LESS [35] for a practical and efficient implementation of our method. The main challenge of using influence-based methods is gradient storage consumption, despite the Hessian calculation is already omitted in our formulation. Therefore, only gradients of LoRA adapters are calculated. Moreover, the gradients are projected to 8192-dimensional subspace via a fixed random matrix. We

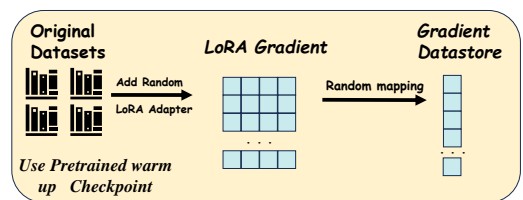 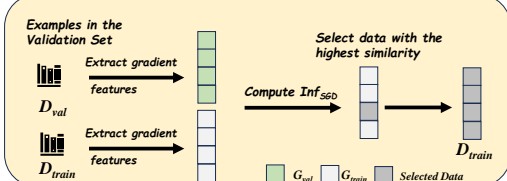

Figure 3: **Data Selection Pipeline for MHR Data.** In step 1, we train a warmup model on the crude dataset to acquire gradient features for both the training and validation sets. In step 2, we match training data candidates with the validation set and select the most influential samples.

note that this leverages the Johnson-Lindenstrauss Lemma [44] that random projection preserves the inner product of gradients, and the recent empirical validation [45] for data selection. The cosine similarity is used instead of the original inner product in common influence definitions because the gradient norm of a language model may have numerical biases [35]. After profiling gradients of the crude training set and the validation set, we rank and retain the top 70% most influential samples for downstream fine-tuning.

**Statistics.** Fig. 1 highlights the advantages and key details of our dataset, which offers the highest resolution among all RS image-text training datasets to date. Fig. 2 visualizes the capability dimensional coverage of both datasets. It can be seen that our datasets span a wider range of task dimensions, showing strong alignment with real-world scenarios. Finally, we summarize the key statistics of our datasets in Tab. 2, including average question/answer lengths, the diversity and quantity of annotated objects, average resolution, and other relevant attributes.

Table 2: **Main statistics of our datasets**

| Statistic | Number |
|---|---|
| Total questions | 81,367 |
|   - SuperRS-VQA | 12,228 |
|   - HighRS-VQA | 69,139 |
| *SuperRS-VQA* | |
|   - Multi-image questions | 150 |
|   - Average question length | 128 |
|   - Average answer length | 22 |
|   - Object category | 18 (Coarse) |
|   - Object size (pixel ratio) | 0.14% |
|   - Average Resolution | 8,376×8,378 |
| *HighRS-VQA* | |
|   - Average question length | 172 |
|   - Average answer length | 41 |
|   - Object category | 60 |
|   - Object size (pixel ratio) | 1.02% |
|   - Average Resolution | 2,000×1,912 |

## 4 Method

Before training on our datasets, we investigate how the Low Semantic Density characteristic, i.e., excessive length visual tokens caused by UHR RS imagery, affects the performance of MLLMs. Specifically, Sections 4.1 and 4.2 separately analyze the low information density of RS imagery from two views: Section 4.1 reveals the redundancy of background tokens and their negative impact on MLLM modeling, while Section 4.2 explores the correlation between scarce objects visual tokens. In Section 4.3, we propose a targeted solution to overcome these challenges.

### 4.1 Overwhelming Background Tokens Hinder MLLM Fine-Tuning on RS Data

*Do background tokens dominate UHR RS imagery?*

In order to learn about a holistic recognition for the background ratio in UHR RS scenes, we conduct preliminary experiments, where the LLaVA-1.5 [33] is employed. Specifically, we randomly selected 100 images from the XLRS-Bench [14], covering natural backgrounds such as sea surfaces, forests, and fields, to provide a diverse set of test samples for the evaluation of background redundancy. Then, we extract 64×64 small images through non-overlapped sliding windows, and sequentially input each into the

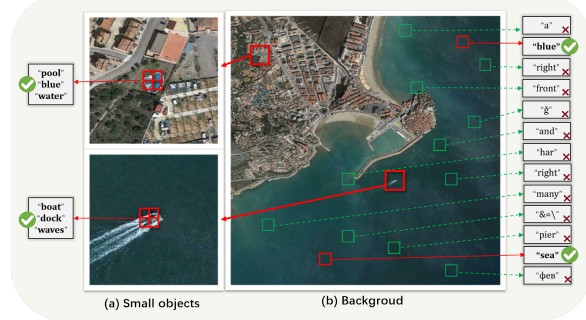

Figure 4: Examples of tokens and their positions that the *logit lens* yields in the late layers.

model. We evaluated the background rate of the UHR RS scene using two methods, following the previous work [46]: **Generative Description**: We prompt the model with "Describe the image" and analyze the generated description to extract key information. **Binary Polling**: We ask the model "Is this image mainly background?" where the background is defined as low-information natural areas (e.g., sea, deserts, vegetation), excluding artificial structures (e.g. buildings, roads, urban areas).

We developed a multi-level semantic parsing framework to precisely quantify MLLM's responses. The framework separates the description from the evaluation content and processes it in five steps: Urban Semantic Recognition, Human-made Structure Analysis, Natural Element Classification, Semantic Categorization, and Probability Gradient Quantification. Details are shown in the appendix. Results show background coverage in RS images reaches up to **73.14%**.

To better understand the semantic information of visual tokens, we use the *logit lens* technique [47]. For each layer, we decode the activation at each token position via unembedding. Details are shown in the appendix. With the *logit lens* technique, we found that in large background areas, only a small portion of the information in the visual tokens is effectively mapped to specific semantic scenes, while most of the background tokens do not align with clear semantic representations (Fig. 4(b)).

> *Do background tokens in RS imagery hinder MLLMs from effectively modeling UHR satellite images?*

Then, we feed the whole UHR RS imagery into MLLM, where the LLaVA-Next-2K [34], a variant of LLaVA-Next [28], is adopted to support higher resolution as far as possible. Initially, we directly perform the inference on XLRS-Bench [14], whose images have an average resolution of 8K×8K. Naturally, the resulting long visual token sequences lead to out-of-memory (OOM) errors, aligning with the first challenge presented in Section 1.

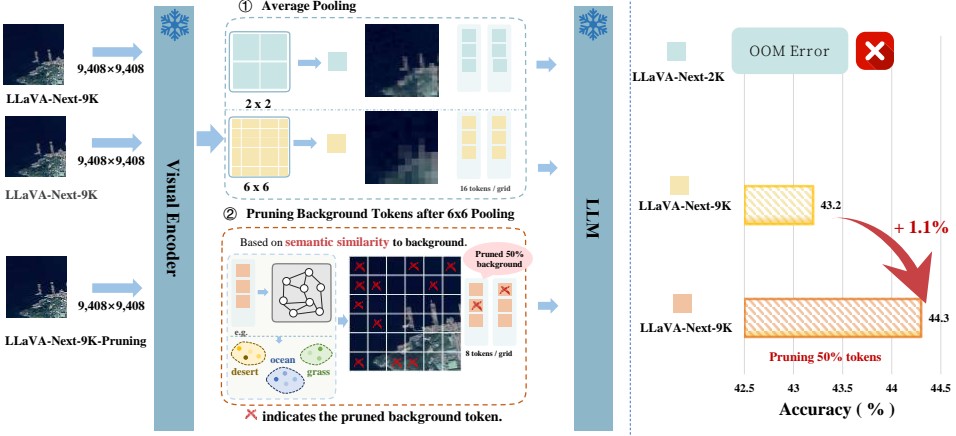

Figure 5: Halving the background tokens by half even resulted in improved performance.

To address the OOM issue, we increase the pooling layer size from the default 2×2 to 6×6, allowing LLaVA-Next-2K to process 9K×9K resolution, called LLaVA-Next-9K. To further reduce redundant background information, we use a background-aware token selection strategy. By calculating semantic similarity between visual tokens and typical background terms (e.g., ocean, desert), we score and rank tokens, then discard the top 50% most redundant ones. The process involves: (1) extracting embedding vectors for typical background terms; (2) calculating each token's highest similarity score with these embeddings as its background score; and (3) ranking tokens by score and removing the top 50%. This approach reduce the background token count. The result is referred to as LLaVA-Next-9K-Pruning.

As Fig. 5 shows, LLaVA-Next-9K underperforms due to its lack of exposure to 9K-resolution training data, emphasizing the need for UHR RS datasets. Surprisingly, despite LLaVA-Next-9K-Pruning uses only half the tokens of LLaVA-Next-9K, it achieves better accuracy. This suggests that excessive background tokens not only introduce additional computational overhead but also impair performance. These findings underscore a critical insight: reducing background redundancy is critical to perform effective modeling under UHR RS images.

From the above two experiments, we confirm that at the token level, RS images exhibit significant background redundancy, ***where key target information constitutes only a small fraction of the image, while most regions lack explicit semantic significance.*** This presents a major challenge to the efficiency of multimodal models.

## 4.2 Scarce Object Tokens Drive MLLM Fine-Tuning on RS Data

After recognizing the background redundancy of UHR RS imagery in visual tokens, we turn to the second key aspect of understanding RS's low semantic density: the localization of critical information.

> *Whether essential information is concentrated in small targets and captured by corresponding visual tokens?*

We conduct ablation experiments to test whether the foreground information is concentrated in specific visual tokens. By removing selected tokens and observing the drop in recognition performance, we assess their importance. The images are sourced from XLRS-Bench, with two pre-processing steps applied to ensure reliability: High signal-to-noise sub-images and Hallucination control. Details are shown in the appendix. Following this procedure, the final dataset includes 1,189 VQA pairs. Fig. 6 provides an overview of our ablation. Details are shown in the appendix.

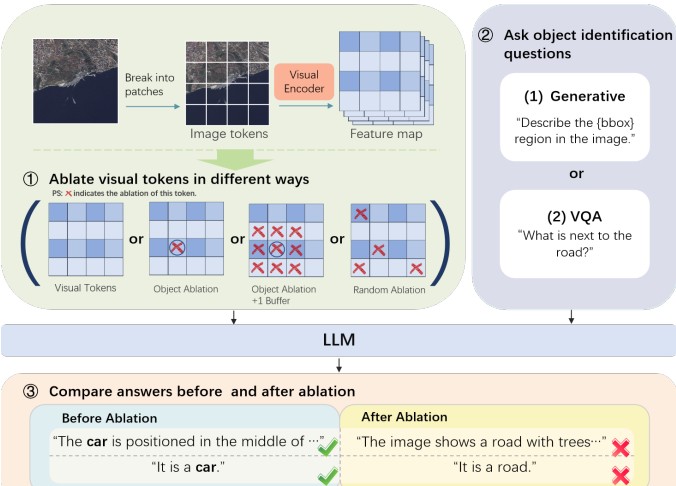

Figure 6: **Overview of object tokens ablation experiments.** ① We ablate some visual tokens that potentially contain object-specific information, ② prompt the model to describe the image, or answer objectspecific questions, then ③ measure the impact of token ablation by calculating the percentage of initially correct object identifications that become incorrect after ablation.

We define the token subset $S$ for ablation using four settings: **(1) Object Tokens**: Tokens aligned with image patches that originally contain the target object; **(2) Object Tokens with Buffer**: Object tokens along with their surrounding tokens. **(3) Register Tokens**: Tokens whose norms deviate by more than two standard deviations from the mean, corresponding to the register tokens identified as encoding global image features [48]; **(4) Random Tokens**: A baseline in which $n$ tokens are randomly ablated.

Tab. 3 shows that ablating object tokens significantly hinders the models ability to recognize targets. Generative decrease and VQA decrease experiments are following the previous work [46]. A larger percentage drop in performance indicates a greater impact of the ablation, meaning the model is more likely to answer incorrectly, thus suggesting that the ablated tokens contain more localized object information. Notably, with a comparable number of ablated tokens, removing object tokens consistently results in a larger performance degradation than random ablation, highlighting the precise localization of object-specific information. $logit$ $lens$ analysis [47] further reveals that visual tokens for scarce objects efficiently converge to accurate semantic

Table 3: **Performance degradation after token ablation.** "Avg. token counts" denotes the average number of tokens ablated per image for LLaVA-1.5.

| Ablation Type (Avg. Token Count) | Generative Decrease (%) | VQA Decrease (%) |
|---|---|---|
| Object (26.5) | 34.9 | 24.8 |
| +1 Buffer (50.2) | 44.8 | 32.6 |
| +2 Buffer (81.8) | 51.1 | 40.6 |
| Register tokens (3.6) | 18.1 | 9.7 |
| Random (5) | 6.7 | 1.1 |
| Random (30) | 14.5 | 4.4 |
| Random (50) | 17.5 | 7.1 |
| Random (80) | 25.6 | 9.9 |

representations (Fig. 4 (a)). Collectively, these findings underscore a key insight in RS: object tokens not only encode critical information but also align closely with actual visual features.

### 4.3 Background Token Pruning and Anchored Token Selection

Through the above analysis, we have identified the Low Semantic Density challenge in UHR RS imagery, characterized by both background and foreground tokens, which poses a significant obstacle in training MLLMs. Based on these findings, we argue that an RS-specific MLLM capable of efficiently handling UHR imagery is both necessary and feasible. As shown in Fig. 7, we propose a two-step token selection strategy for background and object tokens in RS imagery. Specifically, building on our two high-resolution RS datasets: SuperRS-VQA and HighRS-VQA, we SFT existing MLLMs, where we select LLaVA-Next-2K [34], to create RS MLLMs. In practice, we initialize from LLaVA-Next-2K's general-domain pretrained weights.

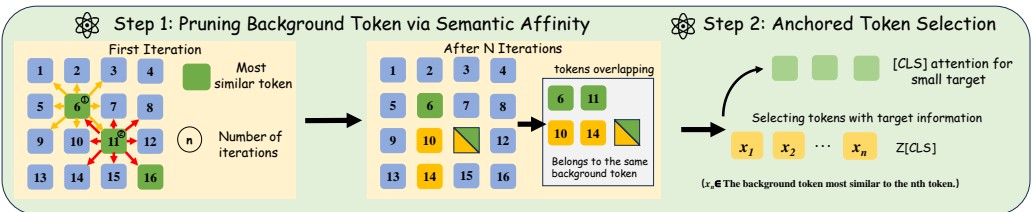

Figure 7: Our two-step method processes visual tokens during MLLMs SFT stage, between the visual encoder and the projection layer.

**Background Token Pruning via Semantic Affinity.** To address the redundancy of background tokens in visual token sequences, we propose an adaptive token clustering strategy for compressing background tokens. Specifically, we construct token-to-token associations by assigning each token $\mathbf{p} = (u, v)$ to a neighboring token $\mathbf{s}$ with probability $q\_\mathbf{s}(\mathbf{p})$. Rather than applying this globally, we restrict associations to the local neighborhood $\mathcal{N}\_\mathbf{p}$, ensuring:

$$\sum_{\mathbf{s} \in \mathcal{N}_\mathbf{p}} q_\mathbf{s}(\mathbf{p}) = 1 \tag{2}$$

After performing this computation once, we iteratively reapply the same process to the selected tokens for $N$ steps, gradually forming an initial cluster. We assume that background tokens, such as those containing ocean features, exhibit high similarity and thus are suited for clustering. We perform this process for all tokens. Note that the clusters formed during later iterations may overlap with those from earlier steps. In such cases, we merge the overlapping clusters into a single one. This adaptive clustering strategy allows us to effectively group similar background tokens, e.g., ocean-related tokens, into one irregular cluster, and forest-related tokens into another, thus compressing redundant visual information more effectively.

**Anchored Token Selection for Scarce Object Retaining.** To prevent small-object or otherwise informative tokens from being lost in a background-oriented pruning stage, we introduce Anchored Token Selection (ATS). ATS leverages the attention map formed between the pretrained ViT's [49] [CLS] token and the remaining image tokens after background pruning. Tokens receiving higher [CLS] to patch attention are deemed more semantically important and are kept, as they likely correspond to informative objects. Specially, the attention map $\boldsymbol{a}_{[\text{CLS}]} \in \mathbb{R}^{1 \times n}$ from the [CLS] token $\boldsymbol{z}_{[\text{CLS}]} \in \mathbb{R}^{1 \times d}$ to other patch tokens $\boldsymbol{Z}_v \in \mathbb{R}^{n \times d}$ is computed by

$$\boldsymbol{a}_{[\text{CLS}]} = \text{Softmax}\left( \frac{\boldsymbol{z}_{[\text{CLS}]} \boldsymbol{W}_Q (\boldsymbol{Z}_v \boldsymbol{W}_V)^T}{\sqrt{d}} \right) = \text{Softmax}\left( \frac{\boldsymbol{q}_{[\text{CLS}]} \boldsymbol{K}_v^T}{\sqrt{d}} \right), \tag{3}$$

where $n$ is the number of remaining image tokens, $d$ is the dimension of hidden states, and $\boldsymbol{W}_Q, \boldsymbol{W}_V$ are the query and key projection matrices of this encoder layer. Note that we utilize the attention map at the second-to-last layer of the visual encoder (*i.e.*, the output layer of CLIP-ViT in LLaVA-Next [28]). Given a compression ratio $r$, we calculate the final number of tokens to retain as $R = n \times r$. Ultimately, we select the top $R$ tokens with the highest attention scores from the attention map $\boldsymbol{a}_{[\text{CLS}]}$, obtaining the final retained tokens $\boldsymbol{Z}_v' \in \mathbb{R}^{R \times d}$. After passing through a multi-modal projector $g$, the remaining image tokens are concatenated with language instructions $\boldsymbol{H}_q$ and fed into the language model with trainable parameters $f_\phi$ to generate the response $\boldsymbol{X}_a$ with $L$ text tokens in the auto-regression manner, where the probability of $\boldsymbol{X}_a$ is computed by $p(\boldsymbol{X}_a|g(\boldsymbol{Z}_v'), \boldsymbol{H}_q) = \prod_{i=1}^{L} f_\phi(\boldsymbol{x}_i|g(\boldsymbol{Z}_v'), \boldsymbol{H}_q, \boldsymbol{X}_{a<i})$.

# 5 Experiments

We perform SFT training on the SuperRS-VQA and HighRS-VQA datasets, with a brief overview of the training details and results in this section. Exploratory and ablation studies are presented in Section 3 to clarify the research motivation. **Additional details, ablation studies of dataset and method, as well as case analyses, are provided in the appendix.**

## 5.1 Main Results

**Experimental Setup.** We use XLRS-Bench [14] for evaluation. The MLLMs evaluated on XLRS-Bench are grouped into three categories: (a) open-source MLLMs; (b) closed-source MLLMs and (c) the specialized RS model. For fair comparison, we used a zero-shot setting with uniform prompts for all MLLMs, including our work. The appendix details the architecture and parameter sizes of each open-source MLLMs, and includes additional results across various settings. Except for GeoChat which was evaluated using its native framework, all other models were evaluated using LMMs-Eval [50, 51]. Following XLRS-Bench [14], we evaluated the accuracy and reported of L-1 dimension for the VQA task, with L-3 and L-4 results available in the appendix.

Table 4: **Experimental results on the perception and reasoning dimensions on XLRS-Bench, with models ranked by average performance.** 'Avg.' represents the average accuracy across sub-tasks. We mark the highest score in red . The full name of sub-tasks can be found in the appendix.

| Method | Perception | | | | | | | | Reasoning | | | | | |
|---|---|---|---|---|---|---|---|---|---|---|---|---|---|---|
| Sub-tasks (L-3 Capability) | OC | RC | OLUC | RLUC | OCC | OCL | OMS | OSR | AD | ECR | RP | RCCD | CCR | Avg. |
| *Remote Sensing MLLMs* | | | | | | | | | | | | | | |
| GeoChat [9] | 16.7 | 29.0 | 2.0 | 23.0 | 21.1 | 16.8 | 35.0 | 24.2 | 33.0 | 43.0 | 10.0 | - | 21.0 | 22.9 |
| *Closed-source MLLMs* | | | | | | | | | | | | | | |
| GPT-4o [12] | 25.0 | 32.0 | 15.0 | 66.0 | 9.5 | 11.3 | 11.7 | 24.6 | 73.0 | 73.0 | 35.0 | 20.0 | 25.0 | 32.4 |
| GPT-4o-mini [52] | 23.3 | 25.0 | 19.0 | 59.5 | 40.9 | 31.0 | 65.0 | 23.6 | 71.0 | 71.0 | 29.0 | 6.7 | 30.0 | 38.1 |
| Claude 3.7 Sonnet [53] | 27.6 | 22.7 | 17.4 | 68.4 | 30.5 | 29.9 | 63.6 | 27.6 | 64.8 | 78.4 | 34.5 | 27.8 | 32.6 | 40.5 |
| Gemini 2.0 Flash [4] | 41.7 | 45.0 | 38.0 | 73.5 | 34.6 | 27.6 | 61.7 | 32.0 | 73.0 | 82.0 | 43.0 | 30.0 | 51.0 | 48.7 |
| *Open-source MLLMs* | | | | | | | | | | | | | | |
| InternLM-XComposer-2.5 [54] | 21.7 | 42.0 | 7.0 | 68.0 | 31.8 | 27.8 | 6.7 | 26.0 | 72.0 | 81.0 | 41.0 | 36.7 | 47.0 | 39.1 |
| LLaVA-Next [28] | 26.7 | 40.0 | 5.0 | 67.0 | 28.8 | 32.8 | 66.7 | 30.0 | 69.0 | 78.0 | 27.0 | 35.0 | 36.0 | 41.7 |
| LLaVA-OneVision-7B [55] | 25.0 | 38.0 | 2.0 | 69.5 | 35.9 | 35.3 | 65.0 | 25.2 | 76.0 | 83.0 | 24.0 | 43.3 | 36.0 | 42.9 |
| InternVL3-8B [56] | 40.0 | 39.0 | 10.0 | 71.5 | 44.5 | 30.8 | 65.0 | 25.2 | 77.0 | 82.0 | 36.0 | 21.7 | 50.0 | 45.6 |
| Qwen2-VL-7B [13] | 26.7 | 40.0 | 11.0 | 73.0 | 35.9 | 34.6 | 61.7 | 31.8 | 70.0 | 81.0 | 35.0 | 46.7 | 48.0 | 45.8 |
| LLaVA-OneVision-72B [55] | 33.3 | 38.0 | 15.0 | 72.5 | 36.3 | 36.3 | 66.7 | 35.6 | 74.0 | 83.0 | 28.0 | 36.7 | 43.0 | 46.0 |
| InternVL2.5-8B [57] | 38.3 | 37.0 | 10.0 | 77.0 | 33.4 | 35.5 | 65.0 | 21.6 | 73.0 | 83.0 | 34.0 | 50.0 | 43.0 | 46.2 |
| Qwen2.5-VL-7B [58] | 33.3 | 40.0 | 31.0 | 77.0 | 33.6 | 35.9 | 66.7 | 36.2 | 68.0 | 72.0 | 27.0 | 38.3 | 45.0 | 47.4 |
| InternVL3-78B [56] | 23.3 | 49.0 | 33.0 | 74.0 | 42.5 | 37.4 | 66.7 | 30.0 | 76.0 | 81.0 | 40.0 | 45.0 | 42.0 | 49.2 |
| Qwen2.5-VL-72B [58] | 33.3 | 47.0 | 39.0 | 80.0 | 45.3 | 42.1 | 65.0 | 34.0 | 71.0 | 74.0 | 37.0 | 43.3 | 42.0 | 50.2 |
| **GeoLLaVA-8K (Our)** | 26.7 | 38.0 | 49.0 | 69.0 | 41.6 | 31.6 | 65.0 | 35.0 | 67.0 | 78.0 | 66.0 | 50.0 | 52.0 | 51.5 |

**Main Results.** After fine-tuning on SuperRS-VQA and HighRS-VQA, our GeoLLaVA-8K delivers outstanding performance across various evaluation tasks. It not only outperforms domain-specific models but also surpasses all existing open- and closed-source models, including the latest Qwen2.5 and InternVL3. Remarkably, with just 7B parameters, GeoLLaVA-8K even outperforms Qwen2.5-VL-72B, the largest and best-performing open-source MLLMs. This impressive gain stems from the high-quality dataset and the targeted compression strategy designed for the low semantic density of RS imagery.

## 5.2 Further Analyses

**Effect of High-Resolution Data vs. Token Optimization Strategies** We conducted an ablation study to analyze the effects of high-resolution data and token optimization strategies. As shown in Table 6, using high-resolution datasets (SuperRS-VQA and HighRS-VQA) already improves performance over the baseline. When combined with Back-

Table 5: **FLOPs and latency of GeoLLaVA-8K under different compression ratios.**

| Compression | Tokens/Grid | Visual Tokens | TFLOPs | Latency (s/img) | Avg. |
|---|---|---|---|---|---|
| 16× (OOM) | – | – | – | – | – |
| 24× | 24 | 14.0k | 198.06 | 2.17 | 51.5 |
| 32× | 18 | 10.5k | 149.08 | 2.03 | 50.3 |
| 48× | 12 | 7.1k | 100.11 | 1.69 | 50.1 |
| 96× | 6 | 3.6k | 51.13 | 1.46 | 49.7 |

ground Token Pruning (BTP) and Anchored Token Selection (ATS), the model achieves the best average accuracy of 51.5%. This shows that while high-resolution data enhances visual understanding, most performance gains come from token optimization, which focuses attention on semantically important regions.

Table 6: **Ablation study on the effect of high-resolution datasets and token optimization strategies.**

| Subtasks | OC | RC | OLUC | RLUC | OCC | OCL | OMS | OSR | AD | ECR | RP | RCCD | CCR | Avg. |
|---|---|---|---|---|---|---|---|---|---|---|---|---|---|---|
| LLaVA-Next-2k (Baseline) | 28.3 | 47.0 | 3.0 | 62.0 | 42.0 | 34.1 | 66.7 | 27.2 | 73.0 | 75.0 | 34.0 | 48.3 | 40.0 | 44.7 |
| + High-Resolution Dataset | 25.9 | 41.0 | 50.0 | 68.0 | 40.4 | 28.5 | 64.9 | 33.4 | 58.0 | 77.0 | 57.0 | 45.0 | 50.0 | 49.1 |
| + Dataset + BTP(OOM) | - | - | - | - | - | - | - | - | - | - | - | - | - | OOM |
| + Dataset + BTP + ATS (GeoLLaVA-8K) | 26.7 | 38.0 | 49.0 | 69.0 | 41.6 | 31.6 | 65.0 | 35.0 | 67.0 | 78.0 | 66.0 | 50.0 | 52.0 | 51.5 |

**Computational Efficiency under Different Compression Ratios** We evaluated the computational efficiency of GeoLLaVA-8K by comparing FLOPs and inference latency across different token compression ratios. As shown in Table 5, higher compression ratios greatly reduce visual tokens and computational cost. At 24× compression, the model processes about 14K tokens with 198.06 TFLOPs and 2.17 s/image latency. Further compression to 96× lowers FLOPs to 51.13 and latency to 1.46 s/image, demonstrating the scalability and efficiency of our token optimization framework.

**Small Object Preservation and CLS Attention under Token Compression.** We analyzed the impact of token compression and CLS-based attention on small-object scenes using a subset of XLRS-Bench defined by COCOs area criteria. Small objects were defined as occupying less than 5% of the image area, and categorized into three levels: Extremely Small (<0.1%), Very Small (0.1%-2.0%), and Normal Small (2.0%-5.0%). A dedicated subset of 1,570 samples covering six tasks (classification, color, motion state, counting, spatial relations) was curated. As shown in Table 7, GeoLLaVA-8K outperformed LLaVA-Next across all tasks, with notable improvements in spatial reasoning, counting, and motion state. These results show that token optimization effectively preserves small-region semantics and reduces information loss. The CLS attention mechanism generally focuses on task-relevant small objects but occasionally favors a single dominant region, suggesting room for further improvement in multi-object focus and attention balance.

Table 7: **Performance comparison on the small-object subset of XLRS-Bench.**

| Dataset | Object Classification | Object Color | Object Motion State | Regional Counting | Overall Counting | Object Spatial Relationship | Overall |
|---|---|---|---|---|---|---|---|
| LLaVA-Next | 38.6 | 29.4 | 62.0 | 35.0 | 22.5 | 20.0 | 34.1 |
| GeoLLaVA-8K | 45.3 | 34.4 | 70.0 | 40.0 | 32.5 | 35.0 | 40.3 |

**Generalization on External Datasets** To evaluate the generalization of GeoLLaVA-8K, we tested it on the new ultra-high-resolution benchmark LRS-VQA [59]. Since LRS-VQA only provides open-ended QA annotations, we converted it to a multiple-choice format using an LLM [12] to generate three plausible distractors per question, followed by structural validation. GeoLLaVA-8K achieved 56.28% accuracy, outperforming LLaVA-Next [28] (55.07%) by 1.21%, confirming strong generalization to new datasets and formats.

Table 8: **Generalization performance on LRS-VQA.**

| Method | Accuracy (%) | Evaluation Format |
|---|---|---|
| LLaVA-Next [28] | 55.07 | MCQ |
| GeoLLaVA-8K (ours) | **56.28** | MCQ |

## 6 Conclusion

In this paper, we addressed two fundamental challenges in scaling vision-language models to UHR RS imagery: the lack of suitable training data and the computational burden of token explosion. To tackle these issues, we introduced two new UHR image-text datasets, SuperRS-VQA and HighRS-VQA, which greatly expand the data available for vision-language tasks on UHR RS images. We also proposed two novel token-efficient strategies: Background Token Pruning and Anchored Token Selection, which effectively reduce the number of visual tokens processed from 8K-resolution images while preserving essential information. Building on these contributions, we developed GeoLLaVA-8K, the first RS-specific MLLM that can directly handle inputs up to 8K×8K resolution. GeoLLaVA-8K achieved state-of-the-art performance on the XLRS-Bench benchmark, outperforming both open- and closed-source MLLMs and demonstrating the effectiveness of our token-efficient approach. These results underscore the value of domain-adapted, token-efficient modeling and provide a new foundation for high-fidelity image understanding in RS.

**Limitation.** We have primarily focused on optical satellite imagery; evaluating GeoLLaVA-8K on other sensor modalities (e.g., synthetic aperture radar or multispectral images) will be important to ensure broader applicability.

**Acknowledgements:** This work was partially supported by the National Natural Science Foundation of China (No. 62372459, No.62376282, No.62225113 and No. 624B2109).

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

# A Appendix

## A.1 Overview of the Appendix

This appendix supplements the proposed **GeoLLaVA-8K** and our datasets (**SuperRS-VQA** and **HighRS-VQA**) with details excluded from the main paper due to space constraints.

The appendix is organized as follows:

- Sec. A.2: More implement details of GeoLLaVA-8K.
- Sec. A.3: The analysis on L-2 capability on GeoLLaVA-8K.
- Sec. A.4: Ablation studys of GeoLLaVA-8K.
- Sec. A.5: More implement details of Pilot Experiments.
- Sec. A.6: Visualizations of samples and challenging cases.
- Sec. A.7: Datasheets for the SuperRS-VQA and HighRS-VQA dataset.
- Sec. A.8: Discussion on limitations and societal impact.

## A.2 More Details of GeoLLaVA-8K

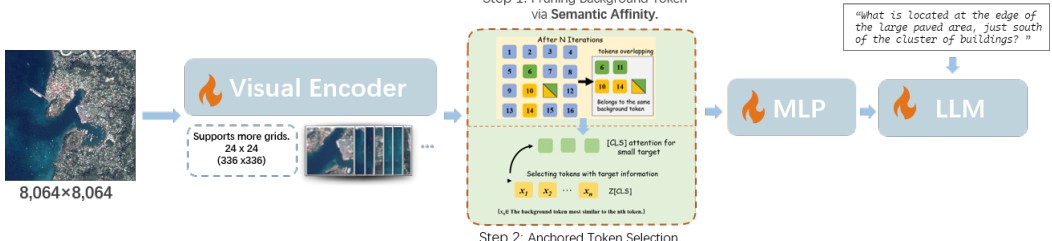

Figure 8: **Overall training framework.** We use SuperRS-VQA and HighRS-VQA datasets for SFT. Our two-step tokens compression method is peformed between visual encoder and projection layer.

GeoLLaVA-8k is developed via full-parameter supervised fine-tuning (SFT) of LLaVA-Next-7B [28] on our SuperRS-VQA and HighRS-VQA datasets (see Fig. 8) We first expanded LLaVA-Next's capacity from processing 7Œ7 visual grids (CLIP's default 336Œ336 pixels per grid, ≈2K resolution) to 24Œ24 grids (≈8K resolution) by directly adjusting the relevant hyperparameters.

To address the explosion of visual tokens, we introduced a two-stage token-efficient mechanism. As described in Section 4.3, the original 2Œ2 pooling was replaced with a clustering-based strategy to merge and compress background tokens. We then applied Anchored Token Selection to further reduce tokens while preserving informative tokens. This approach achieves significantly higher compression: reducing 576 tokens per grid to 24 representative ones, achieving a 24Œ compression rate, which clearly outperforms the 4Œ reduction of original 2Œ2 pooling.

Tab. 9 presents the key training configuration of our GeoLLaVA-8K model. The model is trained on 81K UHR geospatial image-text pairs. We use different learning rates for the visual components (1e-6) and the projection layers interacting with the LLM (5e-6), and optimize the model using ZeRO-2 parallelism with a batch size of 32 for one training epoch.

Table 9: **Training Configuration of GeoLLaVA-8K.**

| Configuration | Parameter |
|---|---|
| Resolution | 8,064×8,064 |
| Dataset | 81,367 (SuperRS-VQA+HighRS-VQA) |
| Batch Size | 16 |
| LR: vision | 1e-6 |
| LR: proj, LLM | 5e-6 |
| ZeRO stage | ZeRO 2 |
| Epoch | 1 |

Tab. 10 presents the detailed structure of XLRS-Bench, the benchmark dataset used to evaluate GeoLLaVA-8K. The dataset is hierarchically organized into two Level-1 categories: Perception and Reasoning, each comprising multiple Level-2 and Level-3 sub-tasks. All Level-3 sub-tasks are formatted as VQA with multiple-choice questions, totaling 3,040 samples. Perception tasks assess

fundamental visual understanding, while reasoning tasks evaluate higher-level cognitive abilities in geospatial contexts.

Table 10: **Characteristics of XLRS-Bench**, used as the benchmark for evaluating GeoLLaVA-8K. The full names of the Level-3 task abbreviations are also provided.

| L1-Task | L2-Task | L3-Task | Abbr. | Annotation Format | Number of Samples | Answer Type |
|---|---|---|---|---|---|---|
| Perception | Counting | Overall Counting | OC | VQA | 60 | Multiple Choice(A/B/C/D) |
| | | Regional Counting | RC | VQA | 100 | Multiple Choice(A/B/C/D) |
| | Scene Classification | Overall Land Use Classification | OLUC | VQA | 100 | Multiple Choice(A/B/C/D) |
| | | Regional Land Use Classification | RLUC | VQA | 200 | Multiple Choice(A/B/C/D) |
| | Object Spatial Relationship | Object Spatial Relationship | OSR | VQA | 500 | Multiple Choice(A/B/C/D) |
| | Object Properties | Object Classification | OCC | VQA | 800 | Multiple Choice(A/B/C/D) |
| | | Object Color | OCL | VQA | 800 | Multiple Choice(A/B/C/D) |
| | | Object Motion State | OMS | VQA | 60 | Multiple Choice(A/B for Yes/No) |
| Reasoning | Route Planning | Route Planning | RP | VQA | 100 | Multiple Choice(A/B/C/D) |
| | Anomaly Reasoning | Anomaly Detection and Interpretation | AD | VQA | 100 | Multiple Choice(A/B/C/D) |
| | Complex Reasoning | Environmental Condition Reasoning | ECR | VQA | 100 | Multiple Choice(A/B/C/D) |
| | | Counting with Complex Reasoning | CCR | VQA | 100 | Multiple Choice(A/B/C/D) |
| | Spatiotemporal Reasoning | Regional Counting with Change Detection | RCCD | VQA | 60 | Multiple Choice(A/B/C/D) |

## A.3 Sub-tasks (L-2 capability) Results on GeoLLaVA-8K

Table 11: Experimental results on the perception and reasoning dimensions of VQA tasks, with models ranked by average performance. 'Avg' represents the average accuracy across sub-tasks.

| Method | Perception | | | | Reasoning | | | |
|---|---|---|---|---|---|---|---|---|
| Sub-tasks (L-2 Capability) | Counting | Scene Classification | Object Spatial Relationship | Object Properties | Planning | Anomaly Reasoning | Complex Reasoning | Spatiotemporal Reasoning |
| *Remote Sensing MLLMs* | | | | | | | | |
| GeoChat [9] | 22.8 | 12.5 | 24.2 | 24.3 | 10.0 | 33.0 | 32.0 | - |
| *Closed-source MLLMs* | | | | | | | | |
| GPT-4o [12] | 28.5 | 40.5 | 24.6 | 10.8 | 35.0 | 73.0 | 49.0 | 20.0 |
| GPT-4o-mini [52] | 24.2 | 39.3 | 23.6 | 45.6 | 29.0 | 71.0 | 50.5 | 6.7 |
| Claude 3.7 Sonnet [53] | 25.2 | 42.9 | 27.6 | 41.3 | 34.5 | 64.8 | 55.5 | 27.8 |
| Gemini 2.0 Flash [4] | 43.3 | 55.8 | 32.0 | 41.3 | 43.0 | 73.0 | 66.5 | 30.0 |
| *Open-source MLLMs* | | | | | | | | |
| InternLM-XComposer-2.5 [54] | 31.8 | 37.5 | 26.0 | 22.1 | 41.0 | 72.0 | 64.0 | 36.7 |
| LLaVA-Next [28] | 33.3 | 36.0 | 30.0 | 42.7 | 27.0 | 69.0 | 57.0 | 35.0 |
| LLaVA-OneVision-7B[55] | 31.5 | 35.8 | 25.2 | 45.4 | 24.0 | 76.0 | 59.5 | 43.3 |
| LLaVA-OneVision-72B[55] | 35.7 | 43.8 | 35.6 | 46.4 | 28.0 | 74.0 | 63.0 | 36.7 |
| InternVL2.5-8B [57] | 37.7 | 43.5 | 21.6 | 44.6 | 34.0 | 73.0 | 63.0 | 50.0 |
| InternVL3-8B [56] | 39.5 | 40.8 | 25.2 | 46.8 | 36.0 | 77.0 | 66.0 | 21.7 |
| Qwen2.5-VL-7B [58] | 33.3 | 42.0 | 31.8 | 44.0 | 35.0 | 70.0 | 64.5 | 46.7 |
| Qwen2.5-VL-72B [58] | 36.7 | 54.0 | 36.2 | 49.3 | 27.0 | 68.0 | 58.5 | 38.3 |
| **GeoLLaVA-8K (Our, 7B)** | 32.3 | 59.0 | 35.0 | 46.1 | 66.0 | 67.0 | 65.0 | 50.0 |

This section highlights the performance of MLLMs across L-2 capabilities, while the related experimental results are shown in Tab. 11.

**Superior Performance in Local Parsing**

GeoLLaVA-8K's superior performance in tasks such as Object Properties (46.1, surpassing the recent Qwen2.5-VL-7B) and Object Spatial Relationship (35.0, only lower than 72B models) demonstrates its ability to effectively parse local details in UHR imagery, laying a solid foundation for comprehensive and accurate ground object understanding. In UHR images, objects often exhibit substantial variations (e.g., buildings or vehicles in different colors), with diverse distribution patterns. GeoLLaVA-8K effectively preserves and leverages these subtle yet critical visual cues under challenging conditions, leading to superior analytical performance.

**Robust Capability in Holistic Understanding.**

GeoLLaVA-8K's top performance in Scene Classification (59.0, 1st) and Complex Reasoning (65.0, tied for 2nd) highlight its ability to capture core semantics and underlying logic from UHR imagery at a macroscopic level. For example, the model can accurately determine a regions overall function, assess the stage of an engineering project, or infer plausible causes of anomalous phenomena by integrating rich visual cues, which is essential for unlocking the full potential of UHR imagery.

**Strong Potential in Dynamic Reasoning** The most distinct advantage of GeoLLaVA-8K lies in its ability to finely comprehend dynamic processes. Its top performance in Planning (66.0, 1st) and

Spatiotemporal Reasoning (50.0, 1st), highlights its effectiveness in modeling long-range dependencies for reasoning. This capability is essential for applications that require precise spatial and temporal understanding, such as urban expansion analysis, disaster response planning, and dynamic monitoring of natural resources, which are challenging for existing models.

## A.4 Ablation studys of GeoLLaVA-8K

Table 12: **Results of ablation experiments on compression rates.** 'Avg.' represents the average accuracy across sub-tasks. We mark the highest score in red .

| Compression Ratio | OC | RC | OLUC | RLUC | OCC | OCL | OMS | OSR | AD | ECR | RP | RCCD | CCR | Avg. |
|---|---|---|---|---|---|---|---|---|---|---|---|---|---|---|
| 16 | | | | | | | | | | | | | | OOM |
| 32 | 28.3 | 39.0 | 46.0 | 65.5 | 42.3 | 34.1 | 65.0 | 35.6 | 64.0 | 74.0 | 63.0 | 46.7 | 50.0 | 50.3 |
| **24 (Our)** | 26.7 | 38.0 | 49.0 | 69.0 | 41.6 | 31.6 | 65.0 | 35.0 | 67.0 | 78.0 | 66.0 | 50.0 | 52.0 | 51.5 |

To investigate the key components of the methodology, we conducted comprehensive ablation studies.

**Visual Token Compression Strategy.** The initial ablation study, with detailed results in Tab. 12, focused on optimizing the visual token compression ratio. We evaluated three settings: 16, 24, and 32. Training with a ratio of 16 frequently led to out-of-memory (OOM) errors, even on multi-GPU setups (8 or 16 GPUs). Although the ratio of 32 used fewer tokens, it resulted in lower accuracy. In contrast, the 24-token setting achieved the best average accuracy (51.5) and enabled stable training on a single node with 8 GPUsstriking an effective balance between performance and efficiency.

Table 13: **Results of ablation experiments on datasets.** 'Avg.' represents the average accuracy across sub-tasks. We mark the highest score in red .

| Datasets | OC | RC | OLUC | RLUC | OCC | OCL | OMS | OSR | AD | ECR | RP | RCCD | CCR | Avg. |
|---|---|---|---|---|---|---|---|---|---|---|---|---|---|---|
| VRSBench-train [20] | 13.3 | 28.0 | 11.0 | 39.5 | 30.1 | 24.6 | 36.7 | 23.6 | 61.0 | 70.0 | 34.0 | 45.0 | 35.0 | 34.8 |
| SuperRS-VQA | 20.0 | 37.0 | 34.0 | 63.0 | 41.4 | 30.3 | 65.0 | 33.4 | 73.0 | 79.0 | 63.0 | 46.7 | 56.0 | 49.4 |
| **SuperRS-VQA + HighRS-VQA** | 26.7 | 38.0 | 48.0 | 69.0 | 41.6 | 31.6 | 65.0 | 35.0 | 67.0 | 78.0 | 66.0 | 50.0 | 52.0 | 51.5 |

**Impact of Training Data Composition.** The second ablation study, summarized in Tab. 13, assessed the contributions of different data sources to the final model performance. We compared three primary scenarios:

   (i) training solely with the VRSBench-train dataset (which contains 123,221 synthetically generated samples);

  (ii) training solely with the SuperRS-VQA dataset (which contains 12,228 high-quality, human-annotated samples);

 (iii) training with a synergistic blend of the SuperRS-VQA dataset and the HighRS-VQA dataset (comprising 12,228 high-quality, human-annotated and 69,139 synthetically generated samples).

As shown in Tab. 13, although VRSBench contains approximately ten times more samples than SuperRS-VQA, training solely on SuperRS-VQA yields significantly better performance (49.4 vs. 34.8). This is likely attributed to the substantial resolution gap: VRSBench averages $512\times512$ pixels, while SuperRS-VQA averages 8,376×8,376, and also reflects the higher annotation quality of SuperRS-VQA. A further accuracy boost to 51.4 was observed when incorporating the HighRS-VQA dataset, validating its complementary value and effectiveness.

## A.5 More implement details of Pilot Experiments.

This section provides more implementation details of the experiments in Section 4.1-4.2.

**Multi-level Semantic Parsing Framework in Sec.4.1.**

The framework separates the description from the evaluation content and processes it in five steps:

**(1) Urban Semantic Recognition**, which detects terms like "city" or "urban" and reduces the background probability if identified;

**(2) Man-made Structure Analysis**, which evaluates semantic relationships between terms like "building," "road," and negation words;

**(3) Natural Element Classification**, which calculates the weight of features like "water" and "forest";

**(4) Semantic Categorization**, which classifies image regions as "Dense," "Sparse," or "None";

**(5) Probability Gradient Quantification**, which assigns values based on semantic strength, such as 20% for urban semantics, 30% for composite man-made structures, and up to 95% for pure natural features.

*logit lens* **Technique in Sec.4.1.** For each hidden state $h_l^i$ at position $i$ in layer $l$, we project it into a probability space of tokenizer vocabulary and select the character with the highest logit. The original paper notes that visual-language models can optimize the alignment between visual and language modalities, enabling visual token representations in deeper layers to decode into language tokens that reflect object semantics. Fig. 9 shows additional examples using the *logit lens* technique to analyze the semantic content of visual tokens.

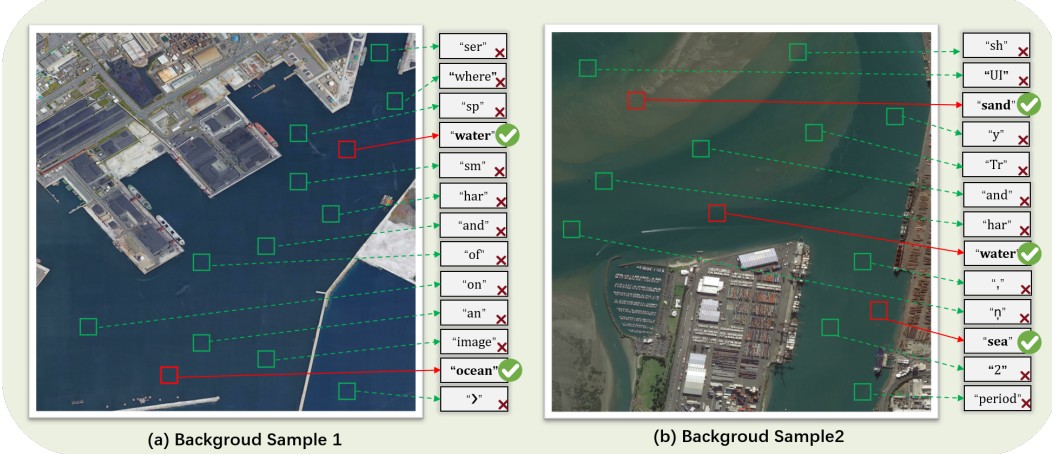

Figure 9: Visual tokens and their semantics detected by the *logit lens* technique.

**Reliability Image Prepare in Sec.4.2.** The images are sourced from XLRS-Bench, with two pre-processing steps applied to ensure reliability:

**(1) High signal-to-noise sub-images**: Following [60], we apply target-centered cropping to increase the object coverage ratio relative to the cropped image size, thereby enhancing target visibility.

**(2) Hallucination control**: Models may infer the presence of objects from context even when visual cues are removed. To counteract this, we generate two versions of each image: the original and a control version with the target masked by noise. We keep only those samples where the model correctly identifies the object in the original but fails in the occluded version, ensuring that recognition relies on visual evidence rather than context.

**Experimental details in Sec.4.2.** Token ablation is performed via substitution. Specifically, token positions corresponding to target objects are identified using XLRS-Bench grounding annotations. These tokens are then replaced with a fixed average embedding, computed as the mean of all visual tokens across the XLRS-Bench images used in this experiment.

We use two methods to assess the impact of token ablation:

1. **Generative Description**: The model is prompted with "Describe the image within the bounding box" using both the original embeddings ($E_A$) and the ablated embeddings ($E_A'$). We compare whether the object $o$ appears in the generated descriptions. If it is mentioned with $E_A$ but not with $E_A'$, the ablated tokens are deemed crucial for recognizing the object.

To ensure the response targets the annotated object, we include the bounding box in the prompt, given the frequent repetition of similar objects in remote sensing images.

2. **Visual Question Answering (VQA)**: We used GPT-4o to generate detailed questions for 400 manually selected remote sensing images, drawn from the XLRS-Bench subset used in this experiment. For example, "What is located at the edge of the large paved area, just south of the cluster of buildings with flat roofs, and adjacent to the narrow green space between the two structures?" All questions were reviewed for clarity and to avoid explicitly naming the target (e.g., "boat"). The models answer is prefixed with "It is a " and we compare its responses before and after ablation.

## A.6 Cases of GeoLLaVA-8K

In this section, we present representative examples from XLRS-Bench in Fig. 10. Each example compares the closed-source Gemini model with the latest open-source models: InterVL-3 and Qwen2.5-VL, across various VQA task formats. Extracting and interpreting objects from large-scale remote sensing imagery remains highly challenging; nonetheless, GeoLLaVA-8K exhibits strong performance in object recognition, localization, and counting under complex, real-world conditions.

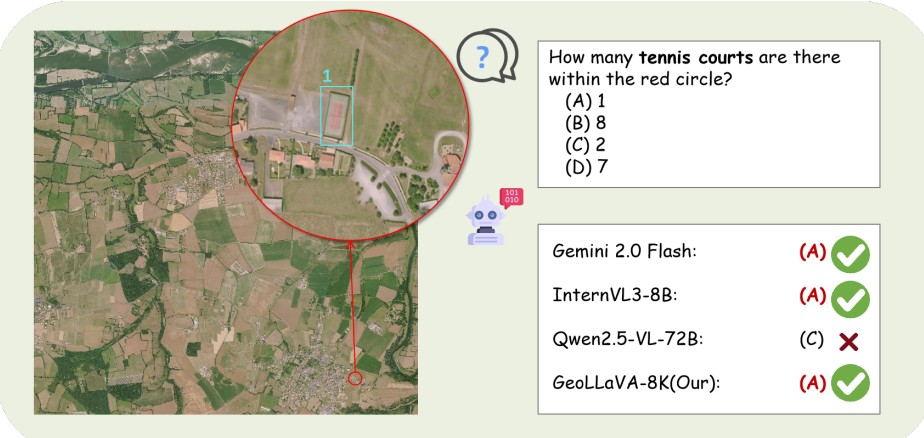

(a) Case of regional counting task.

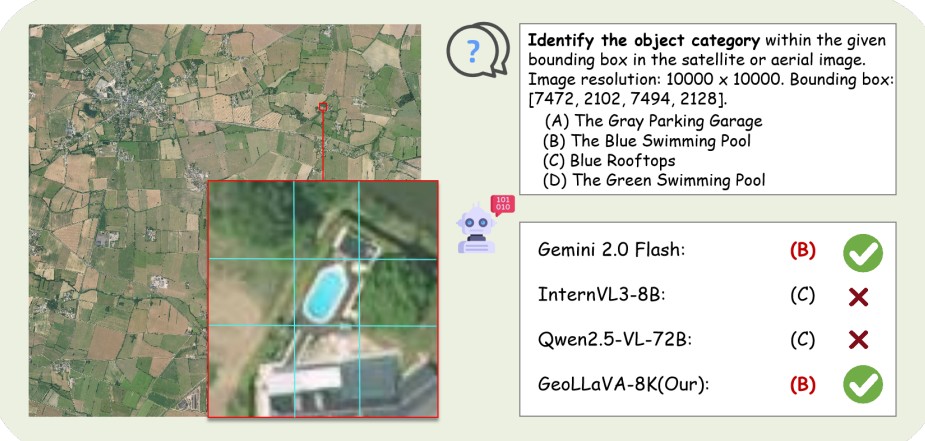

(b) Case of object classification task.

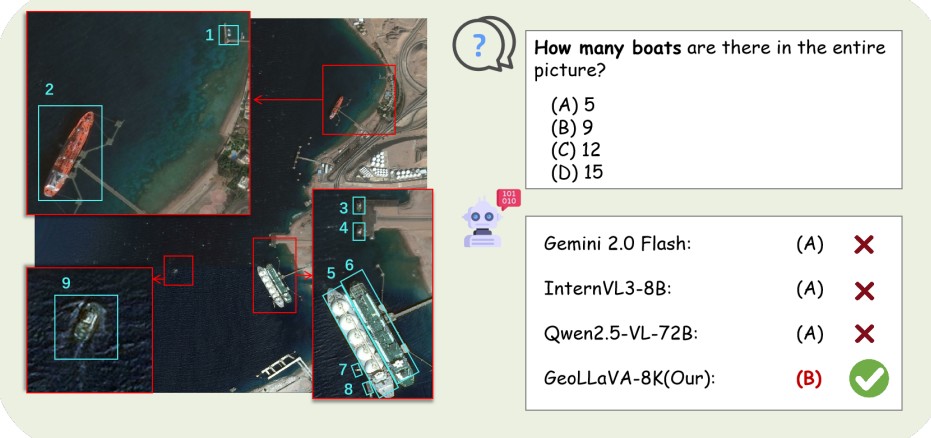

(c) Case of overall counting task.

Figure 10: Cases of Different Task on Various MLLMs (Part 1)

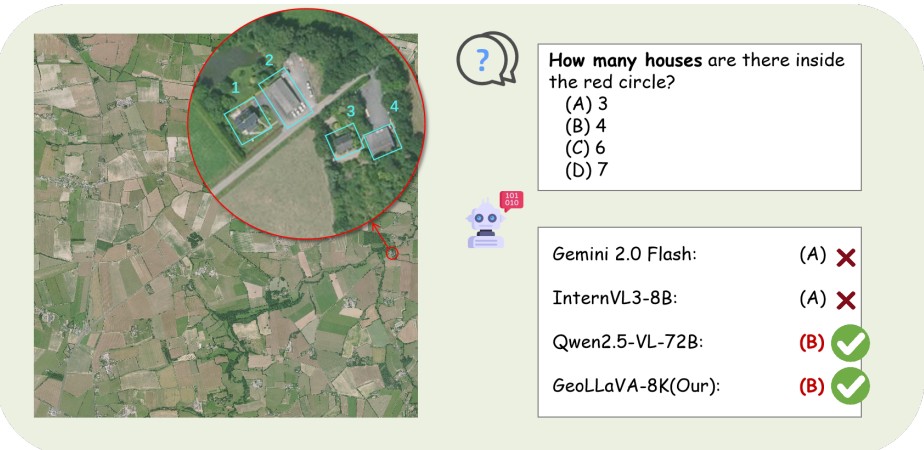

(d) Case of regional counting task.

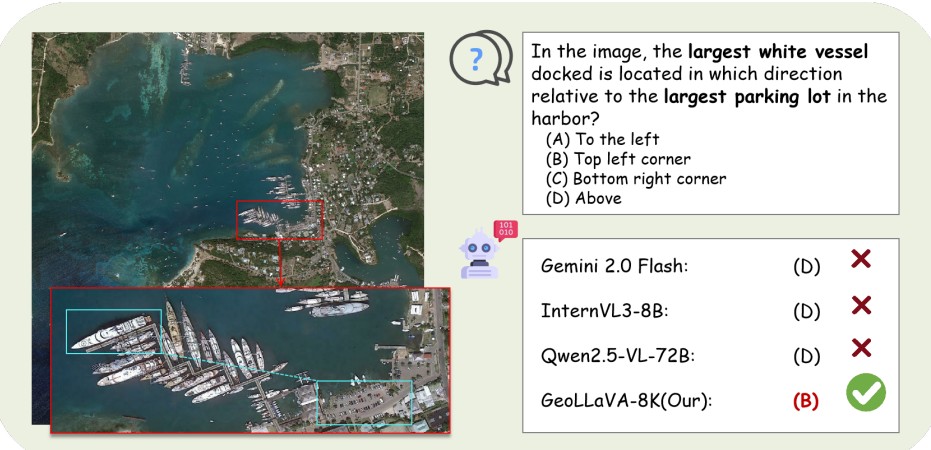

(e) Case of object spatial relationship task.

Figure 10: Cases of Different Task on Various MLLMs (Part 2)

### A.7 Datasheets

In this section, we document essential details about the proposed datasets and benchmarks following the NeurIPS Dataset and Benchmark guidelines and the template provided by Gebru *et al.* [61].

#### A.7.1 Motivation

The questions in this section are primarily intended to encourage dataset creators to clearly articulate their reasons for creating the dataset and to promote transparency about funding interests. The latter may be particularly relevant for datasets created for research purposes.

1. *"For what purpose was the dataset created?"*

   **A:** Ultra-high-resolution (UHR) remote sensing (RS) imagery offers valuable data for Earth observation but pose challenges for existing multimodal foundation models due to the key bottlenecks: limited availability of UHR training data. To address data scarcity, we introduce **SuperRS-VQA** (avg. 8,376×8,376) and **HighRS-VQA** (avg. 2,000×1,912), the highest-resolution vision-language datasets in RS to date, covering 22 real-world dialogue tasks.

2. *"Who created the dataset (e.g., which team, research group) and on behalf of which entity?"*

   **A:** The dataset was created by the following authors:

   • Anonymous authors

3. *"Who funded the creation of the dataset?"*

   **A:** The dataset creation was funded by the affiliations of the authors involved in this work.

#### A.7.2 Composition

Most of the questions in this section are intended to provide dataset consumers with the information they need to make informed decisions about using the dataset for their chosen tasks. Some of the questions are designed to elicit information about compliance with the EUs General Data Protection Regulation (GDPR) or comparable regulations in other jurisdictions. Questions that apply only to datasets that relate to people are grouped together at the end of the section. We recommend taking a broad interpretation of whether a dataset relates to people. For example, any dataset containing text that was written by people relates to people.

1. *"What do the instances that comprise our datasets represent (e.g., documents, photos, people, countries)?"*

   **A:** The dataset primarily consists of ultra-high-resolution remote sensing images captured by satellites, along with their corresponding textual annotations. All datasets utilized in SuperRS-VQA and HighRS-VQA are publicly accessible and nonprofit.

2. *"How many instances are there in total (of each type, if appropriate)?"*

   **A:** SuperRS-VQA and HighRS-VQA includes 81,367 VQA pairs. Details could be found in the Table 2 in main text.

3. *"Does the dataset contain all possible instances or is it a sample (not necessarily random) of instances from a larger set?"*

   **A:** The images in SuperRS-VQA and HighRS-VQA are sourced from existing detection and segmentation datasets, but all textual annotations were independently created by us.

4. *"Is there a label or target associated with each instance?"*

   **A:** Yes, for these ultra-high-resolution images, we have provided VQA pairs instances.

5. *"Is any information missing from individual instances?"*

   **A:** No, each individual instance is complete.

6. *"Are relationships between individual instances made explicit (e.g., users movie ratings, social network links)?"*

   **A:** Yes, the relationship between individual instances is explicit.

7. *"Are there recommended data splits (e.g., training, development/validation, testing)?"*

   **A:** The dataset is designed to train the UHR RS MLLMs.

8. *"Is the dataset self-contained, or does it link to or otherwise rely on external resources (e.g., websites, tweets, other datasets)?"*

    **A:** SuperRS-VQA and HighRS-VQA are self-contained and will be open-sourced on platforms like Hugging Face for easy use.

9. *"Does the dataset contain data that might be considered confidential (e.g., data that is protected by legal privilege or by doctorpatient confidentiality, data that includes the content of individuals non-public communications)?"*

    **A:** No, all data are clearly licensed.

10. *"Does the dataset contain data that, if viewed directly, might be offensive, insulting, threatening, or might otherwise cause anxiety?"*

    **A:** No, SuperRS-VQA and HighRS-VQA do not contain any data with negative information.

### A.7.3 Collection Process

In addition to the goals outlined in the previous section, the questions in this section are designed to elicit information that may help researchers and practitioners create alternative datasets with similar characteristics. Again, questions that apply only to datasets that relate to people are grouped together at the end of the section.

1. *"How was the data associated with each instance acquired?"*

    **A:** The images in SuperRS-VQA and HighRS-VQA are sourced from existing detection and segmentation datasets. We enrich these ultra-high-resolution images with manual annotations, including 81,367 VQA pairs instances. Details are shown in the Section 3 in main text.

2. *"What mechanisms or procedures were used to collect the data (e.g., hardware apparatuses or sensors, manual human curation, software programs, software APIs)?"*

    **A:** We employed professional annotation and quality control teams to complete the annotations for VQA tasks in SuperRS-VQA. For MHR data, we developed a semi-automated annotation. Using task-specific prompts and existing annotations (e.g., bounding boxes in RS detection datasets), we generated text via GPT-4o. We further adopted an influence-based data selection pipeline to improve the relevance of our dataset to UHR downstream tasks and ensure its cultivation of reasoning capabilities for models fine-tuned on it.

3. *"If the dataset is a sample from a larger set, what was the sampling strategy (e.g., deterministic, probabilistic with specific sampling probabilities)?"*

    **A:** Please refer to the details listed in the main text Section 3.

### A.7.4 Preprocessing, Cleaning, and Labeling

The questions in this section are intended to provide dataset consumers with the information they need to determine whether the raw data has been processed in ways that are compatible with their chosen tasks. For example, text that has been converted into a "bag-of-words" is not suitable for tasks involving word order.

1. *"Was any preprocessing/cleaning/labeling of the data done (e.g., discretization or bucketing, tokenization, part-of-speech tagging, SIFT feature extraction, removal of instances, processing of missing values)?"*

    **A:** Yes. To minimize the redundancy of image-text pairs, we deduplicate images within these datasets and remove overlaps with existing benchmark datasets like XLRS-Bench [14].

2. *"Was the 'raw' data saved in addition to the preprocessed/cleaned/labeled data (e.g., to support unanticipated future uses)?"*

    **A:** Yes, raw data is accessible.

3. *"Is the software that was used to preprocess/clean/label the data available?"*

    **A:** Yes, the necessary software used to preprocess and clean the data is publicly available.

### A.7.5 Uses

The questions in this section are intended to encourage dataset creators to reflect on tasks for which the dataset should and should not be used. By explicitly highlighting these tasks, dataset creators can help dataset consumers make informed decisions, thereby avoiding potential risks or harms.

1. *"Has the dataset been used for any tasks already?"*

    **A:** No.

2. *"Is there a repository that links to any or all papers or systems that use the dataset?"*

    **A:** Yes, we will provide such links in the GitHub and the Huggingface repository.

3. *"What (other) tasks could the dataset be used for?"*

    **A:** SuperRS-VQA and HighRS-VQA provide extensive annotations for VQA tasks. It could be used for training the MLLMs.

4. *"Is there anything about the composition of the dataset or the way it was collected and preprocessed/cleaned/labeled that might impact future uses?"*

    **A:** No.

5. *"Are there tasks for which the dataset should not be used?"*

    **A:** N/A.

### A.7.6 Distribution

Dataset creators should provide answers to these questions prior to distributing the dataset either internally within the entity on behalf of which the dataset was created or externally to third parties.

1. *"Will the dataset be distributed to third parties outside of the entity (e.g., company, institution, organization) on behalf of which the dataset was created?"*

    **A:** The datasets will be made publicly accessible to the research community.

2. *"How will the dataset be distributed (e.g., tarball on website, API, GitHub)?"*

    **A:** We will provide uperRS-VQA and HighRS-VQA in the GitHub and the Huggingface repository.

3. *"When will the dataset be distributed?"*

    **A:** We will create a repository to release the data once the paper is officially published.

4. *"Will the dataset be distributed under a copyright or other intellectual property (IP) license, and/or under applicable terms of use (ToU)?"*

    **A:** Yes, the dataset will be released under the Creative Commons Attribution-NonCommercial-ShareAlike 4.0 International License.

5. *"Have any third parties imposed IP-based or other restrictions on the data associated with the instances?"*

    **A:** No.

6. *"Do any export controls or other regulatory restrictions apply to the dataset or to individual instances?"*

    **A:** No.

### A.7.7 Maintenance

As with the questions in the previous section, dataset creators should provide answers to these questions prior to distributing the dataset. The questions in this section are intended to encourage dataset creators to plan for dataset maintenance and communicate this plan to dataset consumers.

1. *"Who will be supporting/hosting/maintaining the dataset?"*

    **A:** The authors of this work serve to support, host, and maintain the datasets.

2. *"How can the owner/curator/manager of the dataset be contacted (e.g., email address)?"*

    **A:** They can be contacted via the email addresses listed on the paper or webpage.

3. *"Is there an erratum?"*

   **A:** There is no explicit erratum; updates and known errors will be specified in future versions.

4. *"Will the dataset be updated (e.g., to correct labeling errors, add new instances, delete instances)?"*

   **A:** Future updates (if any) will be posted on the dataset website.

5. *"Will older versions of the dataset continue to be supported/hosted/maintained?"*

   **A:**

   Yes. This initial release will be updated in the future, with older versions replaced as new updates are posted.

6. *"If others want to extend/augment/build on/contribute to the dataset, is there a mechanism for them to do so?"*

   **A:** Yes, we will provide detailed instructions for future extensions.

## A.8   Limitation and Potential Societal Impact

In this section, we discuss the limitations and potential societal impact of this work.

### A.8.1   Potential Limitations (GeoLLaVA-8K)

- **Scope of Sensors:** GeoLLaVA-8K is tuned for satellite visiblelight imagery; its performance on SAR, multispectral or street-level data is unverified, limiting cross-domain generalisability.

- **Model Scale:** We used only a basic 7B model, while commercial models like Qwen and InternVL have reached 72B. We also aim to scale up model parameters for improved performance.

### A.8.2   Potential Positive Societal Impacts (GeoLLaVA-8K)

- **Fine-grained Environmental Intelligence:** Enables near-real-time QA over 8K imagery for detecting illegal logging, coastal erosion or oil spills, supporting evidence-based policy and SDG monitoring.

- **Disaster-Response Acceleration:** Rapid localisation of collapsed bridges, blocked roads or inundated zones can sharpen rescue logistics and reduce casualty rates.

### A.8.3   Potential Negative Societal Impacts (GeoLLaVA-8K)

- **Surveillance & Dual-Use Risk:** High-precision localisation of vehicles assets lowers the barrier for persistent monitoring and autonomous targeting.

- **Over-reliance on Automated QA:** Persuasive textual answers may be taken at face value; misclassification of disaster extent or land-use could misdirect resources.

### A.8.4   Potential Limitations (SuperRS-VQA & HighRS-VQA)

- **Sensor Homogeneity:** The datasets are dominated by RGB optical satellites; absence of SAR or hyperspectral samples constrains multimodal fusion research.

- **Annotation Consistency:** Crowdsourced VQA answers could contain subtle errors or regional terminology variance, introducing noise in supervision signals.

### A.8.5   Potential Positive Societal Impacts (SuperRS-VQA & HighRS-VQA)

- **Open Benchmark Catalyst:** By releasing the largest-image-size RS VQA corpora (up to 8,376Œ8,376), the datasets establish a transparent yard-stick for future UHR-aware algorithms.

- **Policy-Relevant Insights:** Rich dialogue tasks (22 subtasks) mirror real-world queriese.g., Count illegal fish-farmsproviding a sandbox for developing decision-support tools that aid sustainable development.

### A.8.6 Potential Negative Societal Impacts (SuperRS-VQA & HighRS-VQA)

- **Privacy Concerns:** 153cm GSD tiles can reveal rooftop activity; malicious actors could deanonymise locations or monitor private property.

- **Environmental Footprint of Training:** Fine-tuning large models on tens of thousands of 8K images consumes significant energy; if replicated widely without checkpoint reuse, aggregate carbon emissions rise.

