# OpenReview forum: "GeoLLaVA-8K: Scaling Remote-Sensing Multimodal Large Language Models to 8K Resolution"
_NeurIPS.cc/2025/Conference — NeurIPS 2025 spotlight_

### Official Review · Reviewer_HCHc · 2025-06-30

**Clarity:** 4
**Significance:** 3
**Originality:** 3
**Rating:** 5
**Confidence:** 5

**Summary:**

This paper addresses the key challenges of applying Multimodal Large Language Models (MLLMs) to ultra-high-resolution (UHR) remote sensing imagery. The authors identify two primary bottlenecks: the scarcity of high-resolution training data and the "token explosion" problem, where the large image dimensions lead to computationally prohibitive sequences of visual tokens.
To tackle these challenges, the paper makes three main contributions:
* Data: It introduces two new, large-scale VQA datasets for the RS domain: SuperRS-VQA, a manually annotated dataset with UHR images (avg. 8K resolution), and HighRS-VQA, a larger, semi-automatically generated dataset with medium-to-high resolution images (avg. 2K resolution).
* Method: It proposes a novel two-stage token management strategy to handle UHR inputs efficiently. This involves (i) background token pruning, which uses semantic clustering to merge and remove redundant background tokens (e.g., ocean, forest), and (ii) anchored token selection, which leverages attention from the ViT's [CLS] token to retain the most semantically critical foreground tokens.
* Model: It develops GeoLLaVA-8K, the a remote sensing focused MLLM capable of handling 8K resolution inputs. By integrating the proposed token reduction techniques and fine-tuning on the new datasets, GeoLLaVA-8K achieves new state-of-the-art results on the challenging XLRS-Bench benchmark, outperforming strong open and closed-source models.

**Questions:**

* This work is motivated by efficiency, but the efficiency of the method itself is not measured. Could the authors provide a brief analysis of the computational overhead of your method? A small clarification here would make the practical benefits clearer and provide a more complete picture of the practical trade-offs involved.
* The use of [CLS] attention for selecting important tokens is a key design choice. Could you enhance the appendix with a small qualitative analysis on this point? For example, select 1-2 examples from the XLRS-Bench where the question involves reasoning about multiple small objects or their spatial relationship. Please visualize the [CLS] attention map and the final selected tokens. Does the attention successfully capture all task-relevant objects, or does it sometimes focus on a single, most prominent one?

**Ethical Concerns:**

["NO or VERY MINOR ethics concerns only"]

**Final Justification:**

I maintain my strong recommendation for acceptance (Rating: 5). After reviewing the authors' detailed rebuttal and considering the points raised by the other reviewers, my confidence in the quality and impact of this work has been further solidified. The paper's core strengths include: (1) tackling the critical problem of scaling MLLMs to UHR remote sensing imagery, (2) the contribution of two new high-resolution datasets (SuperRS-VQA, HighRS-VQA), and (3) the token management strategy. My primary concerns, shared in my initial review, included (1) the lack of analysis on the computational overhead of the proposed method and (2) the potential for the [CLS] token-based attention to fail in scenarios with multiple small objects. Both of my concerns were thoroughly addressed by the authors. All in all, i believe the extensive experiments added during the rebuttal have substantially strengthened the paper. I am confident this paper will be a valuable and impactful contribution to the community and strongly advocate for its acceptance.

**Limitations:**

The main paper includes details about the limitations of this work. The authors separately discuss the limitations and impacts for the model (GeoLLaVA-8K) and the datasets (SuperRS-VQA & HighRS-VQA), covering important angles like sensor scope.

**Quality:**

3

**Strengths And Weaknesses:**

Strengths:
* The paper targets a highly significant and difficult problem at the frontier of MLLM capabilities. Scaling models to UHR imagery is a critical next step for many real-world applications, particularly in remote sensing where fine-grained detail is paramount. The work is timely and the problem is very well-motivated.
* The creation and release of SuperRS-VQA and HighRS-VQA are substantial contributions in their own right. Addressing the data scarcity bottleneck with two high-quality, high-resolution datasets will undoubtedly be of great value to the research community. The methodology for creating HighRS-VQA, including the use of an influence-based data selection pipeline (LESS), is sophisticated and demonstrates a commitment to data quality.
* The proposed token reduction strategy is elegantly motivated by a series of pilot studies (Sections 4.1, 4.2) that clearly demonstrate the redundancy of background tokens and the importance of sparse object tokens in RS imagery. The two-step approach of first pruning the background and then selecting the foreground is technically sound.
* The experimental evaluation is comprehensive and convincing. GeoLLaVA-8K's performance on the difficult XLRS-Bench is impressive, setting a new state-of-the-art and outperforming even much larger models like Qwen2.5-VL-72B and proprietary APIs like GPT-4o.
* The paper is well-written, clearly structured, and easy to follow. The figures, particularly Figure 1 (overview), Figure 5 (motivating token pruning), and Figure 7 (method schematic), are excellent aids for understanding the paper's core concepts and contributions.

Weaknesses:
* The "Background Token Pruning" assumes that backgrounds (like forests, water) are semantically homogeneous and can be easily clustered and removed. This might fail in complex, heterogeneous scenes (e.g., suburban areas, sparse agriculture) where the distinction between "background" and "foreground" is blurry, potentially causing important contextual information to be discarded. Moreover, the "Anchored Token Selection" assumes that the [CLS] token's attention is the optimal proxy for semantic importance for all VQA tasks. The [CLS] token is trained for global image classification. For reasoning tasks that depend on the relationship between multiple small but critical objects, the [CLS] token might focus on a single large, prominent object, causing the smaller but contextually vital objects to be pruned. This could be a significant limitation for complex reasoning.
* The method is motivated by reducing the computational burden of UHR images. While it successfully reduces the number of visual tokens fed to the LLM, it introduces a non-trivial pre-processing step involving iterative clustering and attention map computation. The paper provides no analysis of the computational overhead (e.g., latency, FLOPs) of this method itself.

---

> ### Author Rebuttal · Authors · 2025-07-30
>
> ## Q1: This work is motivated by efficiency, but the efficiency of the method itself is not measured. Could the authors provide a brief analysis of the computational overhead of your method? A small clarification here would make the practical benefits clearer and provide a more complete picture of the practical trade-offs involved.
>
> Thanks for your constructive comments. We fully agree that it is necessary to analyse the efficiency. Therefore, we provide inference speed and FLOPs for each compression ratio. We employ the tool calflops and report Tera FLOPs (TFLOPs). We report FLOPs of the LLM side to exclude the computation incurred by the vision encoder, which in our model, accounts for more than a half of the computation. The results of Inference Speed are tested using a image of a resolution of 10,000 $\times$ 10,000 which results in 332k image patches for our 8k model.
> |Compression|Tokens per Grid|Visual Token Volume|TFLOPs of LLM|Inference Speed(second per image)|Avg.|
> |----|---------------|--------------|------|-------|----|
> |16x(OOM)|
> |24x|24|14.0k|198.06|2.17|51.5|
> |32x|18|10.5k|149.08|2.03|50.3|
> |48x|12|7.1k|100.11|1.69|50.1|
> |96x|6|3.6k|51.13|1.46|49.7|
>
> The table presents compression ratios applied consistently during both training and testing—for instance, a 24× ratio is used throughout training and evaluation on XLRS-Bench. Overall, performance declines with higher compression, with the best result—54.5 accuracy—achieved at 24×, the setting adopted by GeoLLaVA-8K. GeoLLaVA-8K supports resolutions up to 8K × 8K, generating a large number of visual tokens. With a 24× compression ratio, about 14K tokens are produced, resulting in higher FLOPs and slower inference. Increasing the ratio to 96× significantly reduces tokens, boosts TFLOPs and inference speed, while maintaining strong performance—49.7 accuracy, outperforming Qwen2.5VL-7B's 47.4. As our focus is model performance, we adopted the 24× setting. In future work, we plan to optimize inference speed and better balance performance and efficiency.
>
>
> ## Q2: The use of [CLS] attention for selecting important tokens is a key design choice. Could you select 1-2 examples from the XLRS-Bench where the question involves reasoning about multiple small objects or their spatial relationship. Does the attention successfully capture all task-relevant objects, or does it sometimes focus on a single, most prominent one?
> We offer comprehensive experiments on small targets to demenstrate that the \[CLS] attention mechanism captures all task-relevant objects in GeoLLaVA-8k, it rarely focuses on the most prominent one. Future work could further improve \[CLS] attention by incorporating fine-grained visual encoders from CLIP-based models such as FG-CLIP or FineCLIP.
>
> Unfortunately, the latest NeurIPS rebuttal policy does not allow image submissions. Therefore, we designed a new experiment to address your concern regarding whether token compression leads to the loss of dense small-object information. We extracted samples that meet the definition of small objects from relevant subtasks in XLRS-Bench to create a new small-object subset of the benchmark. We then evaluated both the baseline LLaVA-Next and our method on this subset. The detailed experimental procedure is outlined below.
>
> We selected six small-object-related tasks from XLRS-Bench—**object classification, object color, motion state, global counting, regional counting, and spatial relation reasoning**. After manual verification, we curated a dedicated small-object subset, which will be open-sourced to support related research.
> * **Small Object Criteria:** COCO dataset defines small objects by absolute pixel area (<32² pixels), which is less suitable for high-resolution remote sensing images. Given COCO's typical image size (400–650 pixels), small objects occupy roughly 2.5%–5% of the image area. Inspired by this, we adopt a relative area-based classification: Extremely Small: <0.1% of image area, Very Small: 0.1%–2.0%, Normal Small: 2.0%–5.0%, Large: ≥5.0%.
> * **For object classification, object color, and motion state tasks**—which include bounding boxes—we calculated the pixel area of each object. We identified 1,450 samples where objects occupy less than 5% of the image. A detailed breakdown is provided in the table below.
>
> |Dataset|Extremely Small (<0.1%)|Very Small (0.1%-2.0%)|Normal Small (2.0%-5.0%)|Total|
> |-|-|-|-|-|
> |Object Classification|531|166|3|700|
> |Object Color|597|101|2|700|
> |Object Motion State|38|12|0|50|
> |Regional Counting|-|37|3|40|
> |Overall Counting|-|38|2|40|
> |Spatial Relationship|-|38|2|40|
> |Overall|1,166|392|12|1,570|
>
> * **For counting and spatial relation tasks** without bounding boxes, we manually annotated targets and calculated their area ratios. Samples with any object exceeding 5% of the image area were excluded, yielding 40 test samples per subtask. Each sample contains only small objects, categorized by the smallest object's area ratio into Very Small(<2.0%) and Normal Small(2.0%–5.0%).
> * **Experimental Results**: GeoLLaVA outperforms the baseline across all six tasks in the newly constructed small-object test set, raising overall accuracy from 34.05% to 40.25% (+6.20%). The most significant gain is in spatial relation reasoning (+15%), followed by global counting (+10%) and motion state (+8%). Color recognition shows the lowest accuracy (34.43%), highlighting the difficulty of extracting color features from small objects. These results suggest that high-level cognitive reasoning tasks hold greater potential for improvement than basic perceptual ones.
>
> |Dataset|Object Classification|Object Color|Object Motion State|Regional Counting|Overall Counting|Object Spatial Relationship|Overall Performance|
> |-|-|-|-|-|-|-|-|
> |LLaVa-Next|38.6|29.4|62.0|35.0|22.5|20.0|34.1|
> |Geollava-8k|45.3|34.4|70.0|40.0|32.5|35.0|40.3|
>
> Overall, GeoLLaVA-8K, trained on SuperRS-VQA and HighRS-VQA, significantly outperforms the baseline LLaVA-Next on small-object tasks, highlighting the substantial benefits of higher resolution(8kx8k). However, as noted in your comment on “information loss in scenarios with dense small targets,” GeoLLaVA-8K still struggles—achieving only \~34% accuracy on some tasks. This indicates limited capability in extracting features from dense small targets, falling short of full recognition and information capture. Addressing this limitation should be a key focus for future research. Thanks for your constructive comments.
>
> **Your feedback is highly constructive and deeply encouraging to us. We sincerely thank you once again. If you have any further questions or concerns, we would be truly grateful for your guidance.**

---

> > ### Comment · Reviewer_HCHc · 2025-08-05
> >
> > I thank the authors for addressing my comments. Reading through the reviews from the other reviewers, I remain confident that this is a good paper that should be accepted.

---

> > > ### Author Response · Authors · 2025-08-05
> > >
> > > Thank you again for recognizing our work. We sincerely appreciate your insightful feedback. Your support is a tremendous source of encouragement for our team.

---

### Official Review · Reviewer_FLPi · 2025-06-30

**Clarity:** 3
**Significance:** 3
**Originality:** 3
**Rating:** 5
**Confidence:** 4

**Summary:**

This presents GeoLLaVA-8K, a multimodal LLM designed for high-resolution remote sensing tasks. The authors introduce SuperRS-VQA (avg. 8,376×8,376) and HighRS-VQA (avg. 2,000×1,912), the highest-resolution vision-language datasets in RS to date, covering 22 real-world 7 dialogue tasks. The authors also propose two strategies: Background Token Pruning and Anchored Token Selection, to reduce the memory footprint caused by the high resolution while preserving key semantics. Ablation studies demonstrate the effectiveness of the design. Experiments show that GeoLLaVA-8K achieves the best performance compared to closed-source or open-source models.

**Questions:**

- Similar to the weakness, it would be helpful to show a comparison with other datasets.
- What is the difference between the tasks that GeoLLaVA-8K performs better or worse, and what could be the potential reason?

**Ethical Concerns:**

["NO or VERY MINOR ethics concerns only"]

**Final Justification:**

Thank you, authors, for addressing all my questions. I will maintain my already positive score.

**Limitations:**

Yes.

**Paper Formatting Concerns:**

No formatting concerns.

**Quality:**

3

**Strengths And Weaknesses:**

**Strengths:**

- The datasets presented in the paper have a large number of samples and cover diverse tasks, which can be very helpful for future research.
- The proposed designs for the model are well-motivated by the features of RS data (e.g., a large fraction of background regions) and lead to better performance. Background Token Pruning and Anchored Token Selection could also be easy to implement.
- The paper writing is clear and easy to understand.

**Weaknesses:**

- The comparison with other models is only conducted on the XLRS-Bench, while preliminary experiments for the designs are also performed on the same benchmark. More datasets for evaluation are needed to show the generalizability of the method.
- Only certain types of tasks show performance benefit, like RP, CCR, and OLUC. But on tasks like RC, AD, and OC, the performance is actually not superior.

---

> ### Author Rebuttal · Authors · 2025-07-30
>
> ## Q1: Show comparison with other datasets.
> We agree that more benchmark datasets should be included. At the time of submission to NeurlPS'25, UHR remote sensing MLLM benchmarks were scarce, with XLRS-Bench being the only open-source, formally published option.
>
> During the rebuttal phase, and following your suggestion, we identified LRS-VQA—a newly accepted ICCV 2025 benchmark—well-suited for evaluating MLLMs on ultra-high-resolution remote sensing tasks. However, LRS-VQA only offers open-ended QA formats, which are incompatible with standard evaluation platforms like VLMEval.
>
> To address this, we created a multiple-choice version of the LRS-VQA dataset and shared it with the LRS-VQA authors and VLMEval platform to support broader community evaluation. Our processing pipeline for LRS-VQA is as follows:
>
> 1. **Plausible Distractor Generation:** We employed a large language model (GPT-4o) to generate three semantically plausible distractors for each question. The prompt explicitly instructed the model to ensure that the distractors:
>    - Are relevant to the question context,
>    - Belong to the same semantic category as the correct answer (e.g., vehicle types, object status),
>    - Do not overlap with the correct answer.
> 2. **Binary Question Handling:** For binary or contrastive questions (e.g., “rural or urban”, “yes or no”), we automatically restricted the number of answer choices to two, ensuring alignment with the question format and avoiding unnecessary ambiguity.
> 3. **Strict Format Enforcement and Post-validation:** We enforced a strict JSON structure during generation, requiring each output to contain:
>    - The original question,
>    - Exactly two or four answer choices labeled with keys "A", "B", (and optionally "C", "D"),
>    - A valid `answer` key pointing to the correct choice.
>       All generated outputs were further post-processed to validate their structural consistency and ensure that the correct answer was included and accurately labeled.
>
> As a result, we successfully generated 7,145 MCQ-style QA pairs out of the 7,333 original questions. This allowed us to perform additional quantitative evaluations using a multiple-choice format that complements the original open-ended benchmark.
>
> We evaluated LLaVA-Next and our GeoLLaVA-8K on LRS-VQA, with **GeoLLaVA-8K achieving 56.28% accuracy—1.21% higher than the LLaVA-Next baseline which achieving 55.07%**—demonstrating our generalization. Looking forward, our data generation pipeline which was used for generating HighRS-VQA can be readily adapted to augment LRS-VQA, further boosting GeoLLaVA-8K’s performance. We will open-source the data generation pipeline to support the community in generating high-quality training data for ultra-high-resolution remote sensing tasks.
>
>
> ## Q2:What is the difference between the tasks that GeoLLaVA-8K performs better or worse, and what could be the potential reason?
>
> Due to space constraints, the main text lacks detailed subtask analysis. Therefore, we present the following analysis:
>
> |Subtasks|OC|RC|OLUC|RLUC|OCC|OCL|OMS|OSR|AD|ECR|RP|RCCD|CCR|Avg.|
> |-|-|-|-|-|-|-|-|-|-|-|-|-|-|-|
> |LLaVA-Next-2k (baseline)|28.3|47.0|3.0|62.0|42.0|34.1|66.7|27.2|73.0|75.0|34.0|  48.3|40.0|44.7|
> |Baseline+SuperRS-VQA and HighRS-VQA|25.9|41.0|50.0|68.0|40.4|28.5|64.9|33.4|58.0|77.0| 57.0|45.0|50.0|49.1|
> |Baseline+SuperRS-VQA and HighRS-VQA + two token pruning strategies|26.7|38.0|49.0|69.0|41.6|31.6|65.0|35.0|67.0|78.0|66.0|50.0|52.0|51.5|
>
> Most subtasks benefit from our datasets and methods, though performance declines are observed in Overall Counting, Regional Counting, Object Color, Object Motion State, and Anomaly Detection，comparing with baseline. These drops stem from the datasets, which lack sufficient or relevant samples for these subtasks. Our proposed method helps alleviate this issue. For example, the Regional Counting task in XLRS-Bench requires counting small objects within red circles—a scenario rarely found in our SFT data. Addressing this requires generating samples that improve the model’s ability to follow such instructions to detect red circles and better capture small-object features.
>
> While extending the GeoLLaVA-8k to five compression ratios, we discovered an intriguing finding: **Several underperforming subtasks (OC and RC) in GeoLLaVA-8K showed significant improvement with higher compression ratios.**
>
> |Subtasks|OC|RC|OLUC|RLUC|OCC|OCL|OMS|OSR|AD|ECR|RP|RCCD|CCR|Avg.|
> |-|-|-|-|-|-|-|-|-|-|-|-|-|-|-|
> |LLaVA-Next-2k (baseline)|28.3|47.0|3.0|62.0|42.0|34.1|66.7|27.2|73.0|75.0|34.0| 48.3|40.0|44.7|
> |24x|26.7|38.0|49.0|69.0|41.6|31.6|65.0|35.0|67.0|78.0|66.0|50.0|52.0|51.5|
> |32x|28.3|39.0|46.0|65.5|42.3|34.1|65.0|35.6|64.0|74.0|63.0|46.7|50.0|50.3|
> |48x|30.0|41.0|46.0|65.0|44.3|31.3|65.0|36.2|64.0|75.0|60.0|44.0|50.0|50.1|
> |96x|43.3|44.0|35.0|62.0|42.1|23.9|65.0|37.2|63.0|74.0|61.0|43.3|52.0|49.7|
>
> The table shows that compression ratios were applied consistently during training and testing. Overall performance declines with higher compression, with the best accuracy—54.5—achieved at 24×, the setting used in GeoLLaVA-8K. Interestingly, some subtasks improve with increased compression. We believe different RS tasks have distinct optimal compression levels—OC, RC, OCC, OCL, and OSR each exhibit unique preferences. Notably, OC and RC achieve substantial gains at 96× compression, outperforming the baseline (LLaVA-Next). This suggests that **downstream tasks may benefit from tailored compression ratio**. For instance, in OC and RC, higher compression may remove redundant information, helping the model focus more effectively on targets.
> As GeoLLaVA-8K is a unified model, we use a single global compression ratio (24×) for evaluation. These results will be included in future versions. We encourage future research to explore task-specific compression strategies, a promising direction for impactful standalone work.
>
> **We sincerely thank you again for your constructive feedback. If you have any further questions or suggestions, we would be more than happy to address them.**

---

> ### Comment · Reviewer_FLPi · 2025-08-05
>
> Thank you, authors, for addressing all my questions. I will maintain my already positive score.

---

> ### Author Response · Authors · 2025-08-06
>
> We greatly appreciate your constructive feedback and meticulous review. We’re pleased to share a brief new about the Q1: the LRS-VQA authors appreciated our newly added Q1 experiments and plan to update their website with our provided MCQ version. Thank you again for your valuable feedback to the RS UHR MLLM community.
> In summary, we would like to express our sincere gratitude for your positive score and further support towards our work!

---

### Official Review · Reviewer_43pb · 2025-07-02

**Clarity:** 3
**Significance:** 3
**Originality:** 2
**Rating:** 4
**Confidence:** 3

**Summary:**

This paper introduces GeoLLaVA-8K, a remote sensing (RS)-specific multimodal large language model designed to address two key challenges in ultra-high-resolution (UHR) RS imagery: scarce training data and token explosion. The authors first construct SuperRS-VQA (8K resolution) and HighRS-VQA (2K resolution), two datasets covering 22 real-world RS tasks, providing high-resolution data for model training. Observing that RS images feature redundant background tokens and sparse target tokens, they propose Background Token Pruning to remove redundant background information and Anchored Token Selection to preserve target-related tokens, reducing computational overhead while maintaining key semantics. Experiments show that GeoLLaVA-8K achieves state-of-the-art (SOTA) performance on the XLRS-Bench benchmark, outperforming all existing models in 8K-resolution RS tasks.

**Questions:**

1. Could the authors supplement detailed ablation results in the main text to clarify the individual contributions of increasing high-resolution data, Background Token Pruning, and Anchored Token Selection strategies? For example, how does removing each strategy alone affect model accuracy on XLRS-Bench?
2. Could the authors provide experimental data on different token compression ratios to demonstrate how compression ratios influence both computational efficiency and semantic retention? Are there optimal ratios for different RS tasks?
3. Was a control experiment conducted to compare models trained solely with high-resolution datasets versus those using the token compression strategies? Please clarify whether performance improvements primarily derive from data resolution or strategy efficiency.
4. Could the authors analyze specific failure cases where GeoLLaVA-8K underperforms baseline models (e.g., LLaVA-Next in certain sub-tasks) to identify whether token compression causes information loss in scenarios with dense small targets or complex backgrounds?

**Ethical Concerns:**

["NO or VERY MINOR ethics concerns only"]

**Final Justification:**

After carefully reviewing the response, The authors' explanations have further strengthened the paper's arguments and solidified my confidence in the contribution and rigor of this work. Therefore, I am happy to maintain my initial positive score.

**Limitations:**

yes

**Quality:**

3

**Strengths And Weaknesses:**

**Strengths**:
1. High-Resolution Datasets：SuperRS-VQA and HighRS-VQA fill the gap in UHR RS training data, offering 10× higher resolution than previous datasets. Data quality is enhanced via influence function-based selection, aligning with real-world RS scenarios.
2. Efficient Token Compression：Background pruning reduces 50% of redundant background tokens, while Anchored Token Selection leverages attention mechanisms to retain target-relevant tokens, enabling 8K processing without memory overflow and improving computational efficiency.
3. Superior Performance：With only 7B parameters, GeoLLaVA-8K outperforms 72B closed-source models, excelling in complex tasks like object counting and route planning, which validates the effectiveness of domain-adapted modeling.
4. Clear Research Logic：The paper presents a coherent research pipeline, from problem identification to dataset construction and strategy design, ensuring accessibility for readers.
5. Open-Source Reproducibility: By committing to release datasets (SuperRS-VQA/HighRS-VQA) and code, the research provides a standardized framework for the RS community. This fosters reproducibility, with 7B-parameter model training procedures detailed sufficiently for replication using 16 A100 GPUs.

**Weaknesses:**
1. Incomplete Experimental Validation：While supplementary materials include ablation studies, the main text lacks detailed descriptions of experimental procedures, especially insufficient ablation analyses that hinder the completeness of findings.
2. Experiments on token compression ratios are insufficient, making it difficult to quantify their impact on overall performance.
3. Lack of comparisons between models trained solely with high-resolution data versus those using both data and compression strategies, limiting clarity on strategy contributions.
4. In certain scenarios (e.g., specific sub-tasks in Table 4), GeoLLaVA-8K underperforms baseline models like LLaVA-Next, suggesting potential negative impacts of token compression strategies in niche contexts.

---

> ### Author Rebuttal · Authors · 2025-07-30
>
> ## Q1:More ablation results
> We fully agree with your suggestion. The ablation studies in appendix, along with the additional experiments and deeper analysis completed during the rebuttal, should be incorporated into the main text. As your recommended, we evaluated the baseline model LLaVA-Next on our high-resolution datasets (SuperRS-VQA and HighRS-VQA).
>
> **Experimental Setup:** We followed LLaVA-Next’s original settings and used the same hyperparameters as in the GeoLLaVA-8K experiments (epoch=1, lr=5e-6, Vision\_tower\_lr=1e-6, warmup\_ratio=0.1, etc.). 'Dataset' means SuperRS-VQA and HighRS-VQA, 'BTP' means the background token pruning method, 'ATS' means anchor token selection method.
> |Subtasks|OC|RC|OLUC|RLUC|OCC|OCL|OMS|OSR|AD|ECR|RP|RCCD|CCR|Avg.|
> |-|-|-|-|-|-|-|-|-|-|-|-|-|-|-|
> |LLaVA-Next-2k (**Baseline**)|28.3|47.0|3.0|62.0|42.0|34.1|66.7|27.2|73.0|75.0|34.0|  48.3|40.0|44.7|
> |Baseline+Dataset|25.9|41.0|50.0|68.0|40.4|28.5|64.9|33.4|58.0|77.0| 57.0|45.0|50.0|49.1|
> |Baseline+Dataset+BTP ||||||||||||||OOM|
> |Baseline+Dataset+BTP+ATS(**GeoLLaVA-8k**)|26.7|38.0|49.0|69.0|41.6|31.6|65.0|35.0|67.0|78.0|66.0|50.0|52.0|51.5|
>
> The results show that while training the baseline model on HighRS-VQA and SuperRS-VQA improves performance, further gains are achieved by adding our BTP and ATS strategies. Most subtasks benefit from our datasets and methods, though performance declines are observed in Overall Counting, Regional Counting, Object Color, Object Motion State, and Anomaly Detection.These drops stem from the datasets themselves, which lack sufficient or relevant samples for these tasks. Our method helps alleviate this issue. For example, the Regional Counting task in XLRS-Bench requires counting small objects within red circles—a scenario rarely found in our SFT data. Addressing this requires generating samples that improve the model’s ability to follow such instructions to detect red circles and better capture small-object features.
>
> **The background token pruning and anchor token selection strategies are strongly coupled.** The first strategy alone causes OOM issues, prompting the need for the second. As an adaptive, iterative method with learnable parameters, the background token pruning struggles to handle numerous background tokens in early training, improving only in later training stages. Therefore, we will conduct additional ablation studies to examine their joint performance under varying compression ratios in the following responce.
> ## Q2:Experiments on different token compression ratios
> The appendix includes an ablation study on compression ratios, but only at three levels. We extended the study to five compression ratios, reporting both computational efficiency and semantic preservation (accuracy). We also detail subtask-level performance across these ratios with comprehensive discussion.
> |Subtasks|OC|RC|OLUC|RLUC|OCC|OCL|OMS|OSR|AD|ECR|RP|RCCD|CCR|Avg.|
> |-|-|-|-|-|-|-|-|-|-|-|-|-|-|-|
> |16x(OOM)|
> |24x|26.7|38.0|49.0|69.0|41.6|31.6|65.0|35.0|67.0|78.0|66.0|50.0|52.0|51.5|
> |32x|28.3|39.0|46.0|65.5|42.3|34.1|65.0|35.6|64.0|74.0|63.0|46.7|50.0|50.3|
> |48x|30.0|41.0|46.0|65.0|44.3|31.3|65.0|36.2|64.0|75.0|60.0|44.0|50.0|50.1|
> |96x|43.3|44.0|35.0|62.0|42.1|23.9|65.0|37.2|63.0|74.0|61.0|43.3|52.0|49.7|
>
> The table presents compression ratios applied consistently during both training and testing. Overall, performance declines with higher compression, with the best result—54.5 accuracy—achieved at 24×. Interestingly, some subtasks improve with higher compression, supporting your commnets *“there are optimal ratios for different RS tasks”*.  Notably, OC and RC see substantial gains under 96× compression, significantly outperforming the baseline (LLaVA-Next). As GeoLLaVA-8K is a unified model, we use a single global compression ratio (24x) for evaluation. We encourage future work to explore task-specific compression strategies, a promising direction for impactful standalone research.
> We also provide inference speed and FLOPs for each compression ratio. We employ the tool calflops and report Tera FLOPs (TFLOPs) of the LLM side to exclude the computation incurred by the vision encoder. The results is tested using a image of a resolution of 10,000 $\times$ 10,000 which results in 332k image patches for our 8k model.
> |Compression|Tokens per Grid|Visual Token Volume|TFLOPs of LLM|Inference Speed(second per image)|Avg.|
> |-|-|-|-|-|-|
> |16x(OOM)|
> |24x|24|14.0k|198.06|2.17|51.5|
> |32x|18|10.5k|149.08|2.03|50.3|
> |48x|12|7.1k|100.11|1.69|50.1|
> |96x|6|3.6k|51.13|1.46|49.7|
>
> GeoLLaVA-8K supports resolutions up to 8K×8K, generating a large number of visual tokens. With a 24× compression ratio, about 14K tokens are produced, resulting in higher FLOPs and slower inference. Increasing the ratio to 96× significantly reduces tokens, boosts TFLOPs and inference speed, while maintaining strong performance—49.7 accuracy, outperforming Qwen2.5VL-7B's 47.4. As our focus is model performance, we adopted the 24× setting. In future work, we plan to optimize inference speed and better balance performance and efficiency.
> ## Q3: Whether performance improvements primarily derive from data resolution or strategy efficiency.
> Your feedback is both constructive and timely. As noted in our response to your first question, we conducted separate experiments on the baseline model using the HighRS-VQA and SuperRS-VQA. The results clearly validate the effectiveness of our datasets and token compression strategies.
> ## Q4: Failure cases to identify whether token compression causes information loss in scenarios with dense small targets?
> Unfortunately, the latest NeurIPS rebuttal policy does not allow image submissions. Therefore, We address this issue from two perspectives, backed by extensive experiments.
>
> **(1) GeoLLaVA-8K does not cause significant loss of small-object information.**
>
> We designed a new experiment to address your concern regarding whether token compression leads to the loss of dense small-object information. We extracted samples that meet the definition of small objects from relevant subtasks in XLRS-Bench to create a new small-object subset of the benchmark. Specially, we selected six small-object-related tasks from XLRS-Bench—**object classification, object color, motion state, global counting, regional counting, and spatial relation reasoning**. After manual verification, we curated a dedicated small-object subset, which will be open-sourced to support related research.
> * **Small Object Criteria:** COCO dataset defines small objects by absolute pixel area (<32² pixels), which is less suitable for UHR RS images. Given COCO's typical image size (400–650 pixels), small objects occupy roughly 2.5%–5% of the image area. Therefore, we adopt a relative area-based classification: Extremely Small: <0.1% of image area, Very Small: 0.1%–2.0%, Normal Small: 2.0%–5.0%, Large: ≥5.0%.
> * **For object classification, object color, and motion state tasks**—which include bounding boxes—we calculated the pixel area of each object. We identified 1,450 samples where objects occupy less than 5% of the image.
>
> * **For counting and spatial relation tasks** without bounding boxes, we manually annotated targets and calculated their area ratios. Samples with any object exceeding 5% of the image area were excluded, yielding 40 test samples per subtask. Each sample contains only small objects, categorized by the smallest object's area ratio into Very Small(<2.0%) and Normal Small(2.0%–5.0%).
>
>
> |Task|Extremely Small (<0.1%)|Very Small (0.1%-2.0%)|Normal Small (2.0%-5.0%)|Total|
> |-|-|-|-|-|
> |Object Classification|531|166|3|700|
> |Object Color|597|101|2|700|
> |Object Motion State|38|12|0|50|
> |Regional Counting|-|37|3|40|
> |Overall Counting|-|38|2|40|
> |Spatial Relationship|-|38|2|40|
> |Overall|1,166|392|12|1,570|
>
> * **Results**: GeoLLaVA-8k outperforms the baseline across all six tasks in the newly constructed small-object test set, raising overall accuracy from 34.05% to 40.25% (+6.20%). The most significant gain is in spatial relation reasoning (+15%), followed by global counting (+10%) and motion state (+8%). Color recognition shows the lowest accuracy (34.43%), highlighting the difficulty of extracting color features from small objects. These results suggest that high-level cognitive reasoning tasks hold greater potential for improvement than basic perceptual ones.
>
> |Dataset|Object Classification|Object Color|Object Motion State|Regional Counting|Overall Counting|Object Spatial Relationship|Overall|
> |-|-|-|-|-|-|-|-|
> |LLaVa-Next|38.6|29.4|62.0|35.0|22.5|20.0|34.1|
> |Geollava-8k|45.3|34.4|70.0|40.0|32.5|35.0|40.3|
>
> Overall, GeoLLaVA-8K significantly outperforms the baseline LLaVA-Next on small-object tasks, highlighting the substantial benefits of higher resolution(8kx8k). However, as noted in your comment on “*information loss in scenarios with dense small targets*,” GeoLLaVA-8K still struggles—achieving only \~34% accuracy on some tasks. This indicates limited capability in extracting features from dense small targets. Addressing this limitation should be a key focus for future research.
>
> **(2) Failure Cases**
>
> In a regional counting task, GeoLLaVA-8K failed to accurately count ships within a red-circled of small harbor. Although 53 ships were present within harbor and 38 within the red circle, the model misinterpreted the instruction, counted the entire harbor, and returned an incorrect result. These errors likely stem from a lack of relevant training samples with red circles.  Addressing this will require generating more targeted samples to enhance the model's ability to follow spatial cues and detect small-object features. While GeoLLaVA-8K generally outperforms the baseline, occasional failure cases—as you pointed out—highlight areas for future improvement.
>
> **All of your suggestions are truly constructive, and our team unanimously agrees that they will greatly improve the quality of our work. We sincerely appreciate again for your valuable guidance.**

---

> > ### Comment · Reviewer_43pb · 2025-08-03
> > **Response to the Author’s Rebuttal**
> >
> > Thank you to the authors for the detailed response, which has addressed my concerns.

---

> > > ### Author Response · Authors · 2025-08-06
> > >
> > > Thank you for taking the time to reply. We are pleased to hear that we have addressed your concerns. If you have any further questions, please let us know promptly so that we can resolve them in the remaining time. We hope you will reconsider our score. Thank you again for your time and positive consideration of our rebuttal.

---

### Decision · Program_Chairs · 2025-09-17

**Decision:**

Accept (spotlight)

**Comment:**

This paper proposes a new high resolution remote-sensing dataset, a new method for training by suppressing irrelevant background tokens, and a new multimodal LLM, GeoLlava. All reviewers find the contributions significant, and any concerns have been addressed by the authors in the rebuttal.